# Epitope binning for multiple antibodies simultaneously using mammalian cell display and DNA sequencing
Ning Lin, Kotaro Miyamoto, Takumi Ogawara, Saki Sakurai, Shinae Kizaka-Kondoh & Tetsuya Kadonosono ✉

Epitope binning, an approach for grouping antibodies based on epitope similarities, is a critical step in antibody drug discovery. However, conventional methods are complex, involving individual antibody production. Here, we established Epitope Binning-seq, an epitope binning platform for simultaneously analyzing multiple antibodies. In this system, epitope similarity between the query antibodies (qAbs) displayed on antigen-expressing cells and a fluorescently labeled reference antibody (rAb) targeting a desired epitope is analyzed by flow cytometry. The qAbs with epitope similar to the rAb can be identified by next-generation sequencing analysis of fluorescence-negative cells. Sensitivity and reliability of this system are confirmed using rAbs, pertuzumab and trastuzumab, which target human epidermal growth factor receptor 2. Epitope Binning-seq enables simultaneous epitope evaluation of 14 qAbs at various abundances in libraries, grouping them into respective epitope bins. This versatile platform is applicable to diverse antibodies and antigens, potentially expediting the identification of clinically useful antibodies.

Monoclonal antibodies, known for their exceptional specificity and high binding affinity for antigenic regions called epitopes, are essential therapeutics for various diseases[1,2]. Epitopes are inherently linked to the functionality of antibodies, as the therapeutic efficacy of an antibody is closely correlated with the epitope on its target[3,4]. Therefore, epitope characterization is a key step in elucidating the functionality of antibodies and evaluating their therapeutic potential.

Epitopes fall into two main categories: linear epitopes, which are defined by a continuous sequence of amino acid residues on antigens, and conformational epitopes, composed of discontinuous amino acid residues that come into proximity through the protein folding of antigens. The antigen residues that constitute the epitopes can be revealed through epitope mapping techniques such as peptide scanning[5-8], site-directed or comprehensive mutagenesis scanning[9-11], X-ray crystallography[12], nuclear magnetic resonance[13], and hydrogen/deuterium exchange[14]. However, these high-resolution analyses are labor-intensive and time-consuming. With an ever-increasing number of antibodies being developed for various targets, a cost-effective and efficient technique is required for epitope characterization of the large array of candidate antibodies, especially in the early stage of antibody discovery. One alternative to epitope mapping is epitope binning, which involves profiling a collection of antibodies and grouping them into distinct bins based on their epitope similarities[15]. Antibodies with similar epitopes often exhibit similar functional characteristics, given the corresponding relationship between epitopes and functionality[3,4]. Even if the precise binding residues remain unrevealed, epitope binning can rationally guide the selection of candidates for further characterization, particularly those sharing overlapping functional epitopes with validated antibodies and exhibiting superior properties.

Several computational approaches have been developed for predicting antibodies with similar epitopes, and these methods are applicable to antibodies featuring both similar sequences and dissimilar sequences[16-19]. However, the limited accuracy and the lack of consideration for epitope variability under physiological conditions significantly restrict their practical application. Experimental epitope binning employs competitive binding assays to assess whether query antibodies (qAbs) target different or overlapping epitopes to reference antibodies (rAbs)[15]. This is accomplished by detecting competitive inhibition of the binding of the qAbs to antigens by the rAbs. Various competitive immunoassay formats (such as classical sandwich, premix, or in-tandem assays) can be used in conjunction with an enzyme-linked immunosorbent assay[20], biolayer interferometry, or surface plasmon resonance[21-24]. Epitope binning can also be achieved by combining a competition strategy with the Luminex multiplex technique[25]. Additionally, flow cytometry (FCM) analysis, using antigen-expressing cells with different fluorescent signatures, enables the determination of competitive

School of Life Science and Technology, Tokyo Institute of Technology, Yokohama 226-8501, Japan.
✉e-mail: tetsuyak@bio.titech.ac.jp

binding profiles and the binning of antibodies[26]. However, all of these approaches require individual production and even purification of qAbs, which limits their application in large-scale evaluations. Therefore, to evaluate multiple antibodies efficiently, it is necessary to develop an epitope binning approach without the use of purified qAbs.

We previously developed a screening system for identifying antigen-binding peptides, namely monoclonal antibody-guided peptide identification and engineering (MAGPIE), in which a peptide library was displayed on the surface of antigen-expressing mammalian cells[27]. In the system, library design and screening for antigen-binding peptides are guided by a validated antigen-binding antibody, called the guide antibody. Because the binding of candidate peptides to the antigen on the cell surface is evaluated by competitive binding with the guide antibody, it is expected that the epitope of the identified antigen-binding peptides should be similar to that of the guide antibody. Based on this principle, we hypothesized that MAGPIE could be applied to evaluate epitope similarity between cell surface-displayed qAbs and an rAb.

In this study, we repurposed MAGPIE to assess whether qAbs and an rAb share a similar epitope. We replaced the peptide library in MAGPIE with qAbs and repurposed the guide antibody to be an rAb. In this way, we developed a parallel epitope binning platform for multiple qAbs named Epitope Binning-seq. Epitope Binning-seq takes advantage of genetically encoded qAbs displayed on the surface of antigen-expressing cells to enable the simultaneous evaluation of epitope similarity for a large number of qAbs by next-generation sequencing (NGS) without the need for individual antibody purification. We first constructed and validated the evaluation system for epitope similarity using an rAb and analyzed the conditions affecting its sensitivity. We then achieved epitope similarity assessment of four qAb-rAb pairs using dual rAbs with distinct epitopes simultaneously. Using the Epitope Binning-seq platform, we successfully evaluated epitope similarity for multiple qAbs in libraries and classified them into different epitope bins based on their enrichment in each rAb group. The effective classification achieved by Epitope Binning-seq demonstrates its great potential as an approach to evaluate the epitope similarity of a large number of qAbs in both natural and artificial libraries.

## Results

### Strategy for evaluating epitope similarity

To assess the epitope similarity of different antibodies, we designed a system to detect competitive binding of antibodies to the same antigen. In this system, a qAb is displayed on antigen-expressing cells in the form of a single-chain variable fragment (scFv) and binds to the antigen on the cell surface (Fig. 1a). A fluorescently labeled rAb is used to specify a target epitope and guide the determination of epitope similarity with qAbs during FCM analysis (Fig. 1b, c). When the rAb is incubated with the cells displaying a qAb, whose epitope is similar to that of rAb, the rAb cannot bind to the antigen because the qAb has already masked the shared epitope (Fig. 1b). By contrast, if the epitopes of rAb and qAb are different, the rAb binds to the antigen on the qAb-displaying cells (Fig. 1c). The binding of rAb to the cells is evaluated by FCM analysis, with rAb-binding [rAb(+)] and rAb-non-binding [rAb(−)] cells being distinguished by fluorescence intensity. Using Epitope Binning-seq, the rAb(−) cell populations are sorted from cell

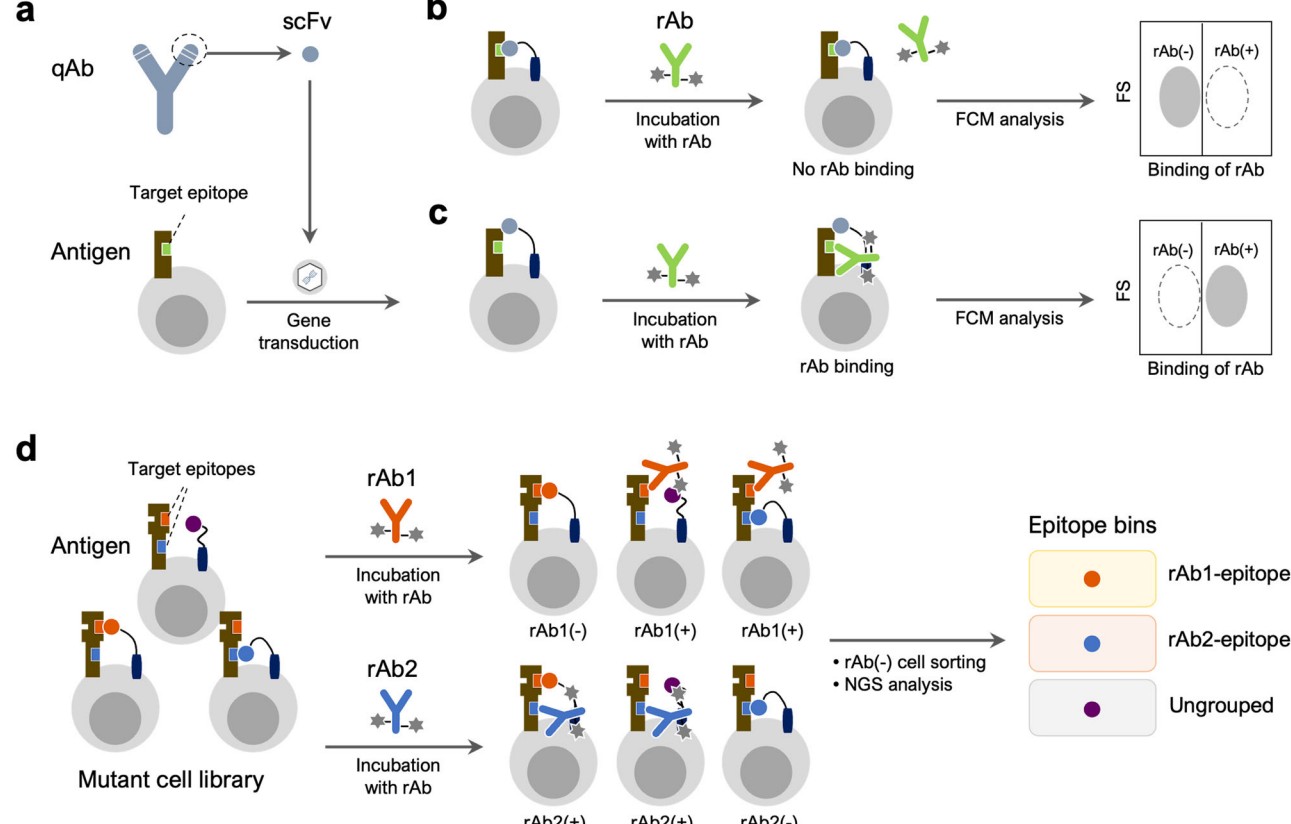

**Fig. 1 | Overview of the evaluation system for epitope similarity and the Epitope Binning-seq platform. a** Cell surface display of the query antibody (qAb). The cDNA encoding a single chain variable fragment (scFv) of qAb is transduced into the antigen-expressing cells. **b** Evaluation of qAbs sharing a similar epitope with the fluorescently labeled reference antibody (rAb). Because the target epitope on the antigen is masked by qAb, the rAb cannot bind to the antigen and an rAb-non-binding [rAb(−)] cell population is detected in FCM analysis. Forward scatter, FS. **c** Evaluation of qAbs with a distinct epitope to that of rAb. The rAb binds to the target epitope and FCM analysis reveals an rAb-binding [rAb(+)] cell population. **d** Parallel antibody classification via Epitope Binning-seq. An antigen-expressing qAb cell library is incubated with rAbs, which bind to distinct epitopes, and each rAb(−) cell population is sorted for NGS analysis to reveal qAb sequences. By analyzing qAbs enriched in the corresponding rAb(−) populations, the qAbs can be classified into different epitope bins, which are composed of qAbs with epitopes similar to each rAb.

libraries comprising various qAbs, their DNA sequences are analyzed by targeted NGS to identify enriched qAbs, and the qAbs expressed by rAb(−) cells are grouped into respective rAb-epitope bins (Fig. 1d).

## System construction and validation

We constructed a model system to evaluate the feasibility of Epitope Binning-seq using the human epidermal growth factor receptor 2 (HER2), which is overexpressed in various cancers[28–31], and HER2-targeting monoclonal antibodies, pertuzumab and trastuzumab, as model antigen and antibodies, respectively. Pertuzumab and trastuzumab are known to bind to HER2 extracellular domain II and IV, respectively [32–34] (Fig. 2a). The binding free energy of residues that comprised the antibody-binding epitopes of HER2 was calculated by molecular dynamics (MD) simulation (Supplementary Fig. 1a). Both pertuzumab- and trastuzumab-binding residues, which had a predicted binding free energy value of <−1.5 kcal/mol, displayed a nonlinear distribution, indicating that these antibodies bind to different conformational epitopes on HER2 (Fig. 2b).

The system requires antigen-expressing cells that display the qAb on the same cell surface. Antigen-presenting K562 cells (K562/HER2), established through lentiviral transduction of K562 cells with a cDNA encoding HER2-T2A-mCherry (Fig. 2c), express the target antigen HER2 on the cell surface and a red fluorescent protein mCherry intracellularly according to the function of a self-cleaving T2A peptide. FCM analysis of K562/HER2 cells with an anti-HER2 antibody revealed a proportional correlation between cell-surface expression of HER2 and expression of mCherry (Fig. 2c). As qAbs, the scFv of pertuzumab, trastuzumab, and a CD25-targeting monoclonal antibody daclizumab, designated scFv(Per), scFv(Tra), and scFv(Dac), respectively, were designed (Supplementary Fig. 2a), and cDNAs allowing each scFv to be displayed on the cell surface as a His-tagged qAb with an intracellular green fluorescent protein sfGFP[27] were constructed (Fig. 2c). After lentiviral transduction of the individual cDNAs into the K562/HER2 cell line, we established qAb-displaying cells, K562/HER2/scFv(Per), K562/HER2/scFv(Tra), and K562/HER2/scFv(Dac) (Supplementary Fig. 2b). A proportional correlation was observed between the amount of cell-surface qAb detected by Alexa Fluor 647 (AF647)-conjugated anti-His tag antibody and the expression of sfGFP in established cell lines (Fig. 2c).

To verify whether the system constructed in this study can evaluate epitope similarity, the three established cell lines were treated with each of the two rAbs, AF647-labeled pertuzumab (Pert-AF647) and AF647-labeled trastuzumab (Tras-AF647). Only K562/HER2/scFv(Per) cells treated with the rAb Pert-AF647 showed a significant rAb(−) population (Fig. 2d), as scFv(Per) masked the epitope for Pert-AF647 on the HER2 cell surface. Similarly, when the three cell lines were treated with Tras-AF647 as the rAb, an rAb(−) cell population was only detected in K562/HER2/scFv(Tra) cells (Fig. 2e). These findings clearly demonstrate that the system works as designed and provide robust evidence of the effectiveness of this system for evaluating epitope similarity.

## Conditions affecting the sensitivity of the evaluation system

Because this evaluation system is based on the competitive binding of qAb and rAb to the antigen, the molecule ratios between qAb and antigen expressed on the cell surface and the concentration of rAb molecules used to treat the cells would affect the sensitivity of the assessment. First, the impact of rAb concentration on evaluation sensitivity was investigated using the model system established above. K562/HER2/scFv(Per) and K562/HER2/scFv(Dac) cells were treated with various concentrations of Pert-AF647 and the percentage of rAb(−) cell population was measured (Fig. 3a). With an rAb concentration as low as 0.1 to 10 nM, the differences in the percentage of the rAb(−) cell population among K562/HER2/scFv(Per) cells were negligible. However, a higher rAb concentration of 100 nM significantly reduced the percentage of the rAb(−) cell population. K562/HER2/scFv(Dac) cells treated with various concentrations of Pert-AF647 showed no observable rAb(−) cell population at any rAb concentrations. Similar results were obtained when Tras-AF647 was used in the same experiment with K562/HER2/scFv(Tra)

cells (Supplementary Fig. 3a). These findings indicate that lower rAb concentrations are advantageous for sensitive evaluation.

Next, the effect of the relative expression levels of qAb and antigen on the evaluation sensitivity was investigated. In the investigation using the model system K562/HER2/scFv(Per) and K562/HER2/scFv(Tra) cells and the corresponding rAbs, Pert-AF647 and Tras-AF647, respectively, we observed rAb(+) cell populations, which would not be expected if the rAb-recognized epitope on the antigen is occupied by the qAb (Fig. 2d, e). We hypothesized that heterogeneous expression levels of HER2 and qAbs in individual cells may cause this phenomenon. For example, when a qAb is expressed at relatively higher levels than HER2, it occupies all available epitopes on HER2, effectively preventing rAb from binding to HER2, whereas when a qAb is expressed at relatively lower levels than HER2, the rAb may be able to bind to HER2. To test this hypothesis, the expression levels of qAb and antigen in rAb(−) and rAb(+) cell populations were estimated according to the expression levels of sfGFP and mCherry, respectively (Fig. 3b and Supplementary Fig. 3b). As expected, rAb(−) cells exhibited higher qAb expression but lower HER2 expression levels compared with rAb(+) cells.

To further confirm the effect of the qAb/antigen balance, we established K562/HER2$^{Lo}$/scFv(Per) and K562/HER2$^{Hi}$/scFv(Per) cells using K562/HER2 cells with low HER2 expression (K562/HER2$^{Lo}$) and high HER2 expression (K562/HER2$^{Hi}$), respectively (Fig. 3c). The ratio of HER2 molecules to qAb molecules for K562/HER2$^{Lo}$/scFv(Per) cells was lower than that of K562/HER2$^{Hi}$/scFv(Per) cells, as evidenced by the higher sfGFP/mCherry ratio (Fig. 3d). When these cells were treated with Pert-AF647, a significant reduction in the rAb(−) cell population was observed in K562/HER2$^{Hi}$/scFv(Per) cells compared with K562/HER2$^{Lo}$/scFv(Per) cells (Fig. 3e). Similar results were obtained with K562/HER2/scFv(Tra) cells reacted with Tras-AF647 (Supplementary Fig. 3c, 3d). Taken together, these results demonstrate that the higher the qAb/antigen ratio, the better the performance of the epitope similarity evaluation system.

## Simultaneous evaluation of epitope similarity using two distinct rAbs

Since we confirmed that the constructed evaluation system works effectively for both rAbs Pert-AF647 and Tras-AF647, we expanded its functionality to allow simultaneous evaluation of epitope similarity using two rAbs with distinct epitopes. To verify the efficacy of simultaneous evaluation, pertuzumab and trastuzumab were labeled with AF647 and Alexa Fluor 488 (AF488) to prepare two rAbs, Pert-AF647 and Tras-AF488, respectively. As a result of the similar emission/excitation wavelengths of AF488 and sfGFP, the qAb cDNA was reconstructed to co-express with blue fluorescent protein (BFP) (Fig. 4a). After lentiviral transduction of the cDNAs into K562/HER2 cells, the resultant cells, namely K562/HER2/scFv(Per)$^B$, K562/HER2/scFv(Tra)$^B$, and K562/HER2/scFv(Dac)$^B$, expressed qAbs on their surfaces and BFP intracellularly (Fig. 4b). These cells were then reacted with two different rAbs, Pert-AF647 and Tras-AF488, and analyzed by FCM. Both rAbs bound to K562/HER2 and K562/HER2/scFv(Dac)$^B$ cells (Fig. 4c), whereas a Pert-AF647-non-binding/Tras-AF488-binding cell population was detected in K562/HER2/scFv(Per)$^B$ cells and a Pert-AF647-binding/Tras-AF488-non-binding cell population was detected in K562/HER2/scFv(Tra)$^B$ cells (Fig. 4c). Furthermore, when the cells displaying different qAbs were mixed pairwise at a 1:1 ratio and evaluated using two rAbs with different epitope recognition, the presence of cells expressing scFv(Per) or scFv(Tra) sharing the same epitope with either Pert-AF647 or Tras-AF488 led to the observation of corresponding populations that exhibited no binding to Pert-AF647 or Tras-AF488 (Fig. 4d). These results suggested that using two distinct rAbs enables the simultaneous evaluation of four rAb-qAb pairs with distinct epitopes in a single experiment.

## Evaluation of qAbs with various HER2-binding affinities

In conventional competitive assays, high-affinity antibodies demonstrate stronger competition than low-affinity antibodies, and as a result, even if low-affinity antibodies target a functional epitope, they may be neglected

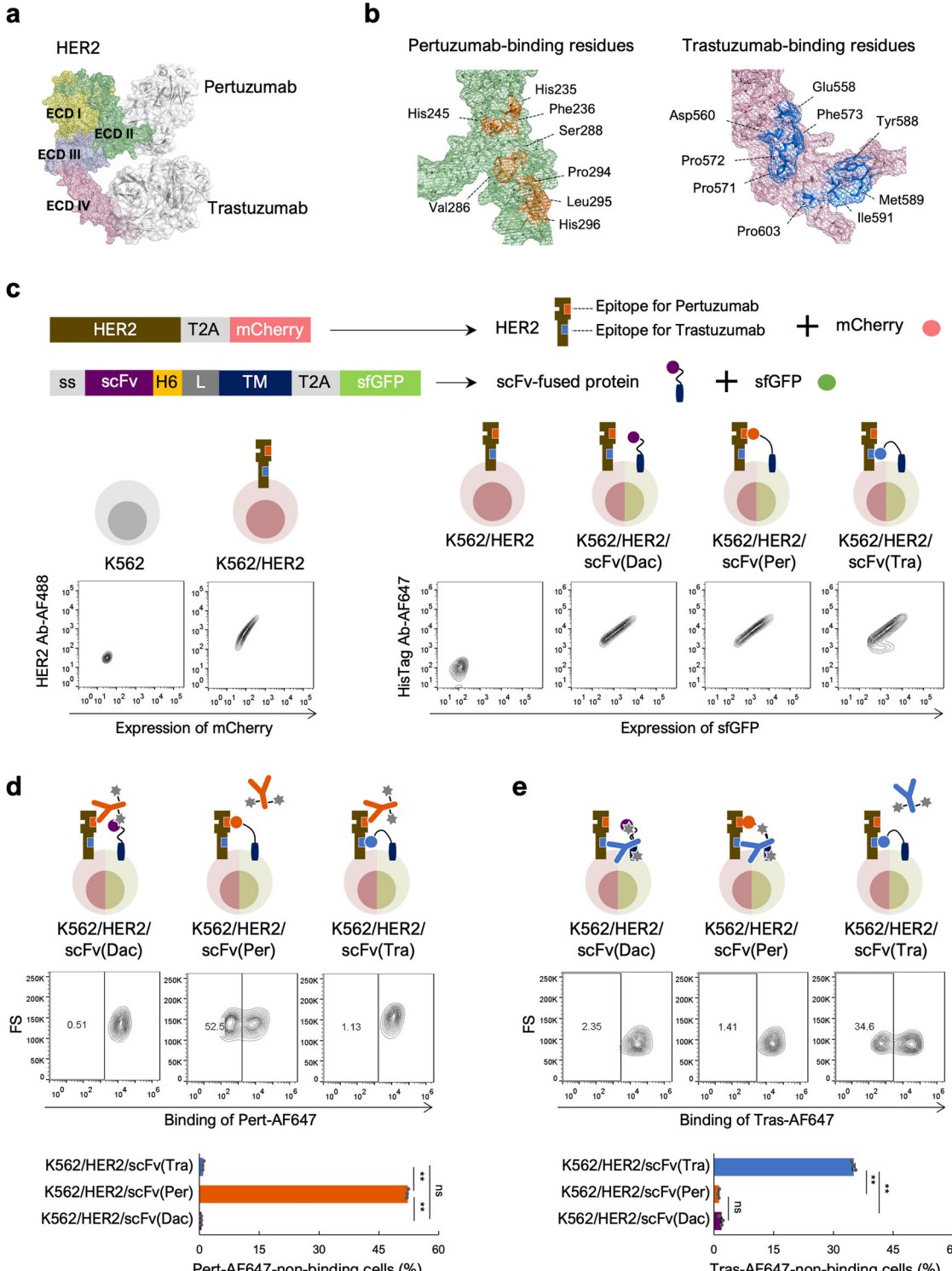

**Fig. 2 | Validation of the evaluation system using HER2-binding antibodies.**
**a** Binding of pertuzumab and trastuzumab to the HER2 extracellular domain (ECD). The model was generated from PDB entries 1N8Z and 1S78. **b** Conformational epitopes on HER2 for pertuzumab and trastuzumab. Antibody-binding residues are shown. **c** Establishment of cell lines displaying qAbs on HER2-expressing cells. Diagram shows cDNA constructs in the plasmids pCSII/HER2-mCherry and pCSII/scFv-sfGFP for expressing HER2 and displaying scFv on the cell surface, respectively. ss, secretion signal peptide; H6, His6 tag peptide; L, linker peptide; TM, transmembrane domain. Expression of HER2 on K562/HER2 cells was analyzed by FCM using Alexa Fluor 488-conjugated anti-HER2 antibody (HER2 Ab-AF488). Surface display of qAbs on K562/HER2/scFv(Dac), K562/HER2/scFv(Per), and K562/scFv(Tra) cells was analyzed by FCM using Alexa Fluor 647-conjugated anti-

His6 tag antibody (HisTag Ab-AF647). **d** Evaluation of epitope similarity between qAb and Pert-AF647. K562/HER2/scFv(Dac), K562/HER2/scFv(Per), and K562/scFv(Tra) cells were incubated with 1 nM Pert-AF647 prior to FCM analysis. Representative contour plots are shown. Percentage of Pert-AF647-non-binding cells calculated using FCM plots from three independent experiments is also shown as the mean ± SEM. **p < 0.0001. ns, not significant (Tukey's multiple comparisons test). n = 3. **e** Evaluation of epitope similarity between qAb and Tras-AF647. K562/HER2/scFv(Dac), K562/HER2/scFv(Per), and K562/scFv(Tra) cells were incubated with 1 nM Tras-AF647 prior to FCM analysis. Representative contour plots are shown. Percentage of Tras-AF647-non-binding cells calculated using FCM plots from three independent experiments is also shown as the mean ± SEM.
**p < 0.0001. ns, not significant (Tukey's multiple comparisons test). n = 3.

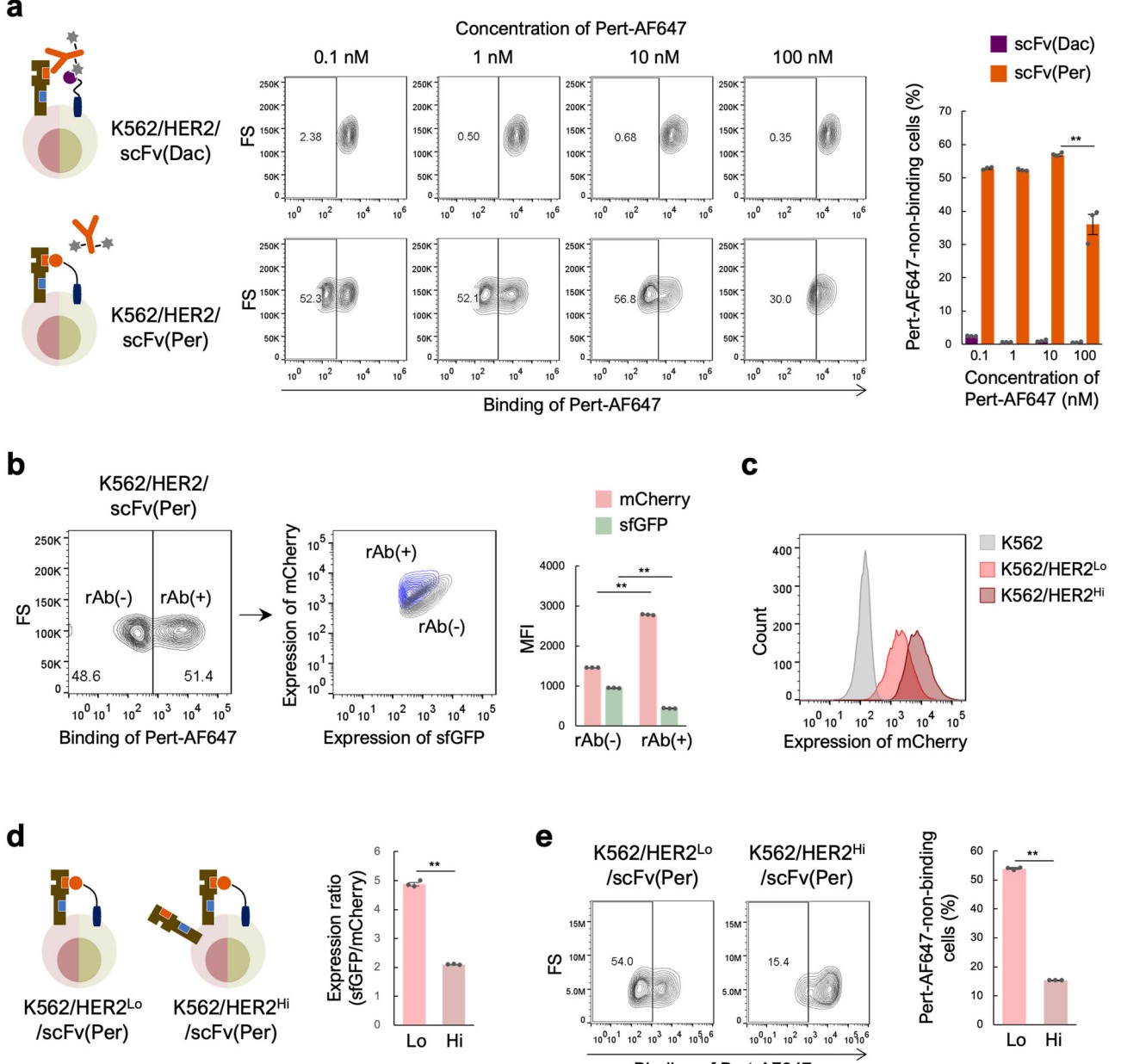

**Fig. 3 | Effects of the Pert-AF647 concentration and the balance between qAb and HER2 expression levels on evaluation sensitivity. a** Evaluation of epitope similarity between qAb and various concentrations of Pert-AF647. K562/HER2/scFv(Dac) and K562/HER2/scFv(Per) cells were incubated with 0.1–100 nM Pert-AF647 prior to FCM analysis. Representative contour plots are shown. Percentage of Pert-AF647-non-binding cells calculated using FCM plots from three independent experiments is also shown as the mean ± SEM. **$p < 0.0001$ (Tukey's multiple comparisons test). n = 3. **b** Expression levels of scFv(Per) and HER2 in Pert-AF647-non-binding [rAb(−)] and Pert-AF647-binding [rAb(+)] cell populations. K562/HER2/scFv(Per) cells were incubated with 1 nM Pert-AF647 prior to FCM analysis and the expression levels of scFv(Per) and HER2 were estimated by the fluorescent intensity of sfGFP and mCherry, respectively. Representative contour plots are shown. Mean fluorescence intensity (MFI) values of sfGFP and mCherry in the rAb(−) and rAb(+) cell populations calculated using FCM plots from three independent experiments are also shown as the mean ± SEM. **$p < 0.0001$ ($t$-test). n = 3.

**c** Expression level of HER2 on K562/HER2$^{Lo}$ and K562/HER2$^{Hi}$ cells, as estimated by the fluorescent intensity of mCherry. Representative histograms are shown. **d** Balance between scFv(Per) and HER2 expression levels in K562/HER2$^{Lo}$/scFv(Per) (Lo) and K562/HER2$^{Hi}$/scFv(Per) (Hi) cells. Expression levels of scFv(Per) and HER2 were estimated by the fluorescent intensity of sfGFP and mCherry on FCM analysis, respectively, and the relative expression ratio was calculated using the MFI values of sfGFP and mCherry from three independent experiments. The mean ± SEM is shown. **$p < 0.0001$ ($t$-test). n = 3. **e** Epitope similarity evaluation between qAb and Pert-AF647 using cells with different HER2 expression levels. K562/HER2$^{Lo}$/scFv(Per) (Lo) and K562/HER2$^{Hi}$/scFv(Per) (Hi) cells were incubated with 1 nM Pert-AF647 prior to FCM analysis. Representative contour plots are shown. Percentage of Pert-AF647-non-binding cells calculated using FCM plots from three independent experiments is also shown as the mean ± SEM. ** $p < 0.0001$ ($t$-test). n = 3.

because of their weaker competitive ability. Therefore, to investigate the ability of our system to evaluate qAbs with weak HER2-binding affinities, mutant scFvs were designed from scFv(Per) and scFv(Tra). In these mutants, based on the results of binding free energy calculations, five consecutive residues contributing to HER2 binding to various degrees were substituted with alanine one-by-one (Supplementary Fig. 1b). Wild-type and mutant scFvs, scFv(Per), N52A mutant scFv(Per), P53A mutant scFv(Per), N54A mutant scFv(Per), S55A mutant scFv(Per), G56A mutant

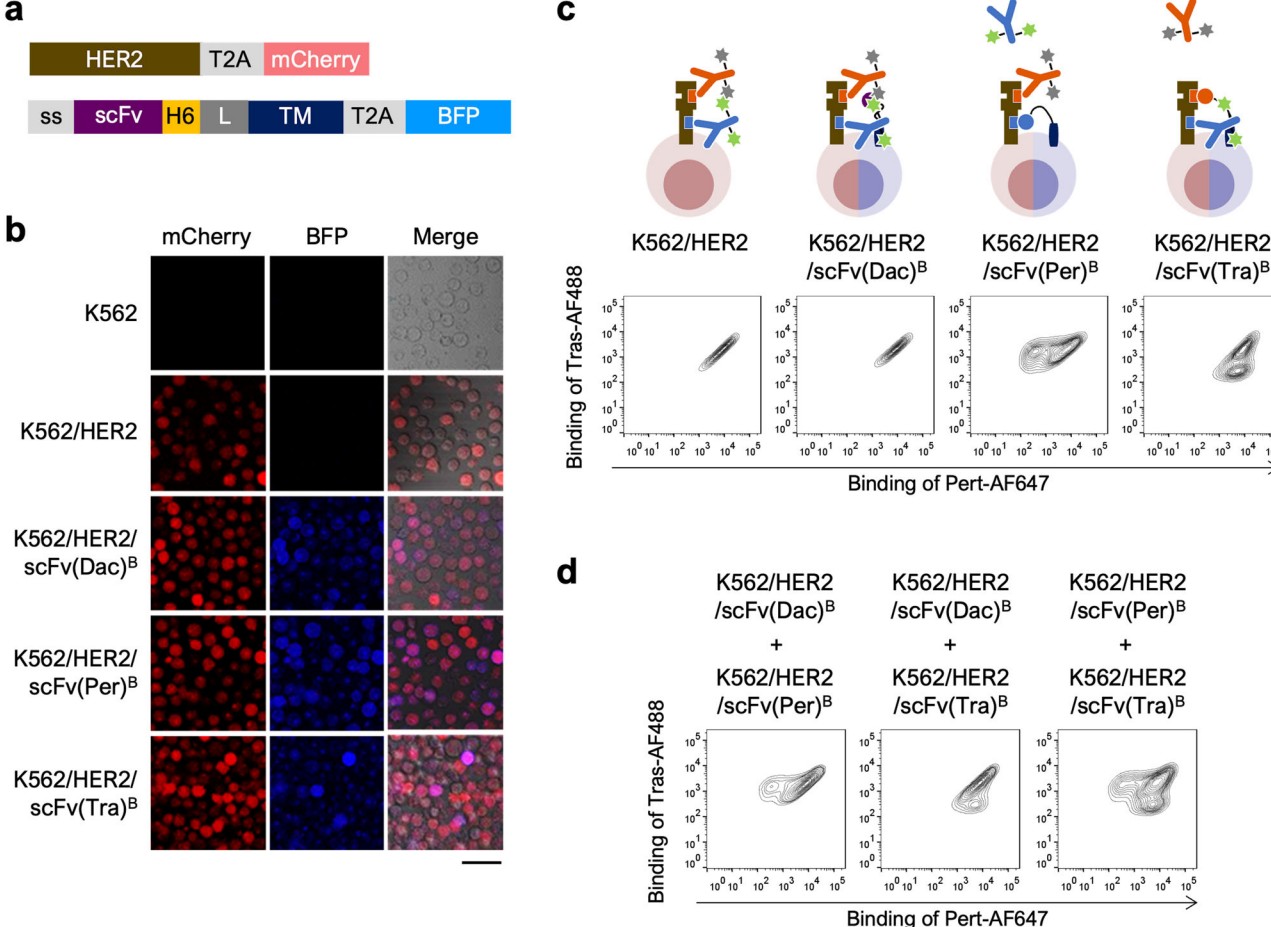

**Fig. 4 | Simultaneous evaluation of epitope similarity using two rAbs. a** Diagram of the cDNA constructs in the plasmids pCSII/HER2-mCherry and pCSII/scFv-BFP for expressing HER2 and displaying scFv on the cell surface, respectively. ss, secretion signal peptide; H6, His6 tag peptide; L, linker peptide; TM, transmembrane domain. **b** mCherry and BFP expression in K562/HER2/scFv(Dac)$^B$, K562/HER2/scFv(Per)$^B$, and K562/HER2/scFv(Tra)$^B$ cells. Scale bar = 50 μm. **c** Epitope similarity evaluation between qAb and two rAbs. K562/HER2/scFv(Dac)$^B$, K562/

HER2/scFv(Per)$^B$, and K562/HER2/scFv(Tra)$^B$ cells were incubated with both 1 nM Pert-AF647 and 1 nM Tras-AF488 prior to FCM analysis. Representative contour plots are shown. **d** Epitope similarity evaluation between two qAbs and two rAbs. K562/HER2/scFv(Dac)$^B$, K562/HER2/scFv(Per)$^B$, and K562/HER2/scFv(Tra)$^B$ cells were mixed pairwise in a 1:1 ratio and incubated with both 1 nM Pert-AF647 and 1 nM Tras-AF488 prior to FCM analysis. Representative contour plots are shown.

scFv(Per), scFv(Tra), G101A mutant scFv(Tra), D102A mutant scFv(Tra), G103A mutant scFv(Tra), F104A mutant scFv(Tra), Y105A mutant scFv(Tra), scFv(Dac), and scFv of PD-1 targeting nivolumab [scFv(Niv)] (Supplementary Fig. 2a) are hereafter referred to as Per, Per-N52A, Per-P53A, Per-N54A, Per-S55A, Per-G56A, Tra, Tra-G101A, Tra-D102A, Tra-G103A, Tra-F104A, Tra-Y105A, Dac, and Niv, respectively. These 14 scFvs were purified as fusion proteins with the single-chain Fc (scFc) domain (Supplementary Fig. 4a) and their binding affinities to HER2-expressing cells were analyzed (Fig. 5a and Supplementary Fig. 4b). Per and Tra showed the strongest binding with the lowest dissociation constant ($K_D$) values, while almost all mutants had weaker binding affinities than the parent scFvs. Dac and Niv exhibited no observable binding.

These scFvs were then displayed on the K562/HER2 cell surface as qAbs and analyzed for rAb-binding by FCM. When cells displaying Per and its mutants were reacted with 0.1 nM Pert-AF647, the percentage of the rAb(−) cell population in Per (52.7%) remained relatively consistent for most Per mutants (49.4–59.0%), except for a substantial reduction for Per-N52A (29.3%), which has the lowest binding affinity (Fig. 5a, b). These results demonstrate that Per and its mutants bind to the same or similar epitopes on HER2. Furthermore, when the concentration of Pert-AF647 was increased to 10 nM, the percentage of the rAb(−) cell population decreased in proportion to the binding affinity of the scFvs (Fig. 5c). Similar results were obtained when analyzing cells displaying Tra and its mutants

with Tras-AF647 (Supplementary Fig. 5a, b). These results clearly indicate that the binding affinity of qAbs affects the sensitivity of epitope evaluation, and that lower rAb concentrations allow for the evaluation of both low-affinity and high-affinity qAbs.

**Epitope Binning-seq for analyzing various qAbs**

In our evaluation system, the cDNA encoding qAb is integrated into the genome of each cell, therefore the amino acid sequence of qAb can be revealed by analyzing the DNA sequence of the cells. Taking advantage of this feature, we constructed an Epitope Binning-seq platform, which classifies large numbers of qAbs in parallel according to their epitope similarity. First, K562/HER2/scFv cells were mixed to generate a cell library displaying 14 scFvs as qAbs. To comprehensively evaluate the performance of the platform, the Per, Per mutants, and Niv were represented in high abundance, while the Tra, Tra mutants, and Dac were represented in low abundance. After reacting the cell library with rAb, the rAb(−) cell population was sorted, and the enrichment ratio, which indicates the ratio of the change in frequency of qAbs in sorted cells compared with the original library, was analyzed using NGS read counts (Fig. 6a, Supplementary Fig. 6a, b). Upon collecting the rAb(−) cell population that reacted with 0.1 nM Pert-AF647, the percentage of cells displaying Per and its mutants increased (Fig. 6b), with all showing a relative enrichment ratio greater than 3 (Fig. 6c), indicating the substantial enrichment of desired qAbs through our epitope

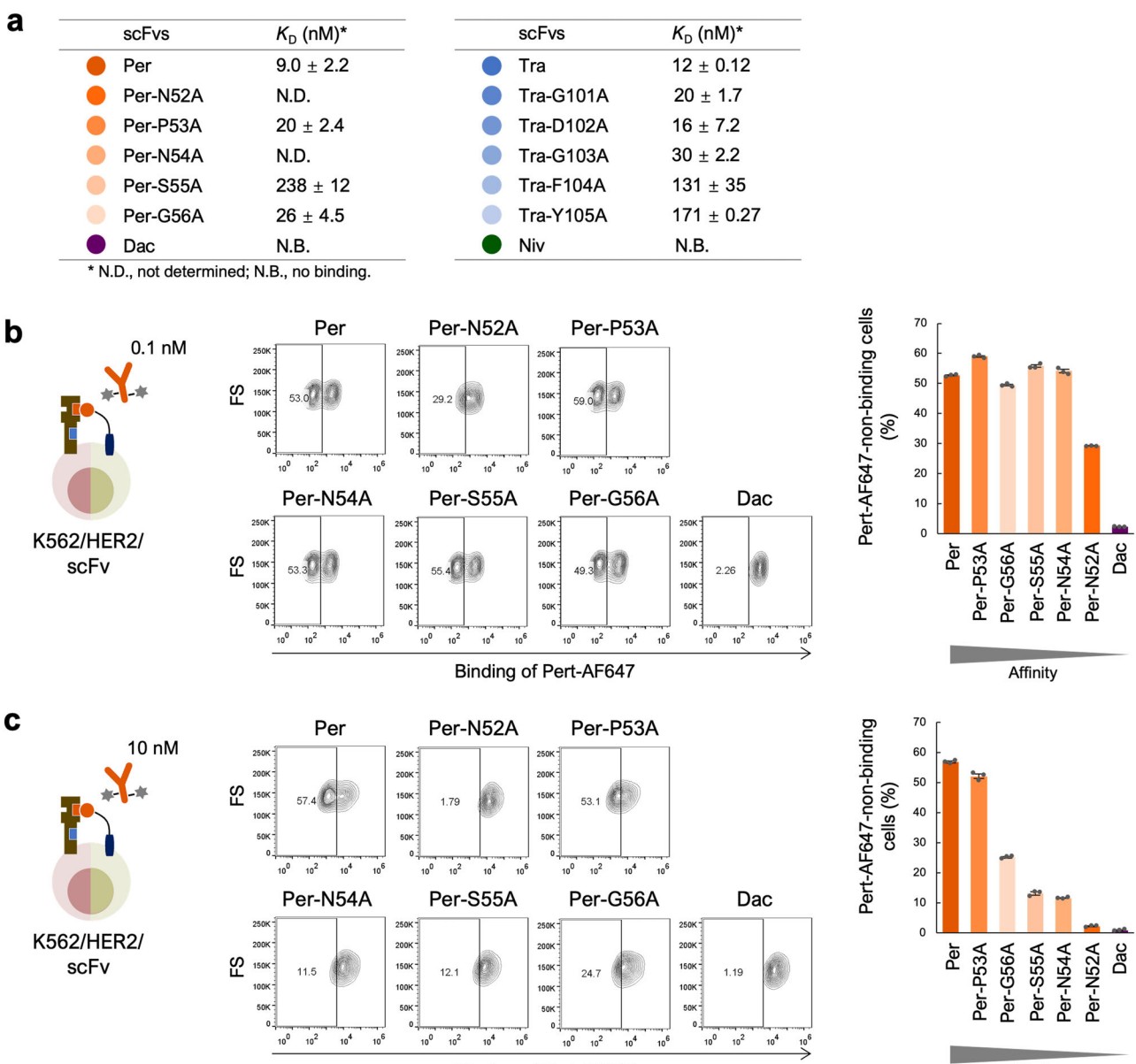

**Fig. 5 | Evaluation of qAbs with various HER2-binding affinities. a** HER2-binding affinity of qAbs. $K_D \pm$ SEM values are shown. n = 3. **b** Epitope similarity evaluation between qAbs and Pert-AF647 at 0.1 nM. K562/HER2 cells displaying Per, Per mutants, and Dac were incubated with 0.1 nM Pert-AF647 prior to FCM analysis. Representative contour plots are shown. Percentage of Pert-AF647-non-binding cells calculated using FCM plots from three independent experiments is also shown as the mean ± SEM. n = 3. **c** Epitope similarity evaluation between qAb and Pert-AF647 at 10 nM. K562/HER2 cells displaying Per, Per mutants, and Dac were incubated with 10 nM Pert-AF647 prior to FCM analysis. Representative contour plots are shown. Percentage of Pert-AF647-non-binding cells calculated using FCM plots from three independent experiments is also shown as the mean ± SEM. n = 3.

binning approach. In contrast, the relative enrichment ratios of Tra and its mutants (which have different epitopes from pertuzumab), and Dac and Niv (which bind to different antigens than HER2), were less than 3. Furthermore, when the reaction concentration of Pert-AF647 was increased to 10 nM, only Per and Per-P53A, which have strong binding affinities, showed a relative enrichment ratio above 3 (Fig. 6d, e). When the rAb was changed to 0.1 nM of Tras-AF647, Tra and its mutants exhibited substantial enrichment, all showing a relative enrichment ratio greater than 3, much greater than those of other qAbs (Supplementary Fig. 7a, b). As expected, the slight enrichment of scFvs with relatively high binding affinity was observed at the rAb concentration of 10 nM (Supplementary Fig. 7c, d). In this study case, through the identification of substantially enriched antibody clones under the guidance of each rAb at the concentration of 0.1 nM, the 14 qAbs were grouped into three distinct epitope bins: pertuzumab-epitope,

trastuzumab-epitope, and ungrouped (Fig. 6f). Specifically, Per and Per mutants, along with Tra and Tra mutants, were classified into the respective pertuzumab- and trastuzumab-epitope groups, which was consistent with their origins, while qAbs Dac and Niv, which exhibited no binding to the antigen, did not belong to either of the two rAb bins. These results indicate that Epitope Binning-seq can effectively classify antibodies with similar epitopes as rAbs among a wide variety of antibodies. Under certain conditions, in addition to the epitope, the relative binding affinity can also be roughly reflected according to the varying responses to different rAb concentrations.

## Discussion

Comparing epitopes between approved and newly developed antibodies is an important process in the development of antibody-based drugs with novel

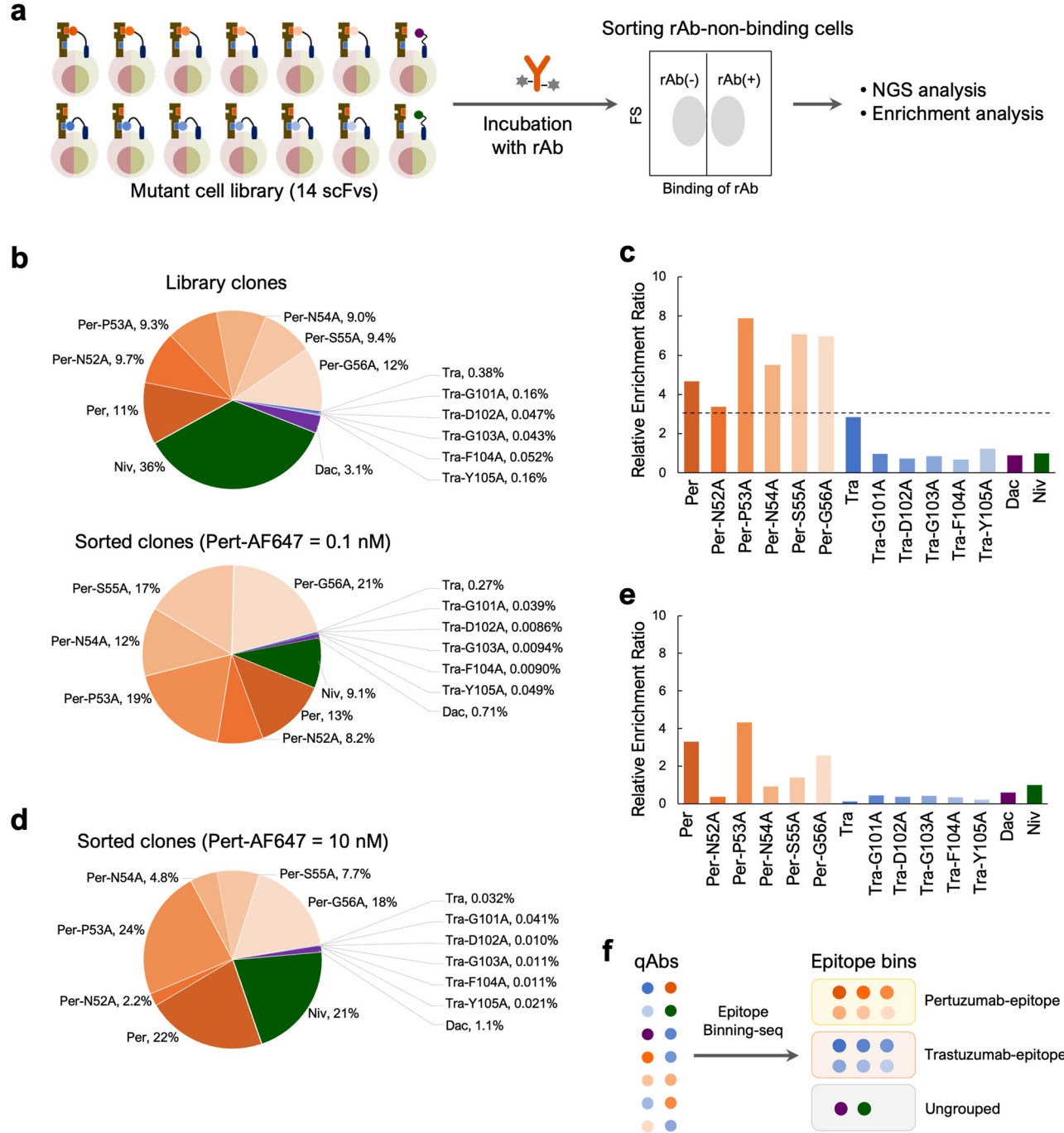

**Fig. 6 | Epitope Binning-seq for analyzing various qAbs. a** Schematic diagram of Epitope Binning-seq. A cell library is generated by mixing K562/HER2 cells expressing 14 different scFvs, followed by incubation with an rAb and a collection of the rAb-non-binding [rAb(−)] cell population using a cell sorter. The genome is extracted from these cells and read counts of each scFv are analyzed by NGS. The enrichment ratio of each scFv is calculated from the occupancy of each sequence before and after sorting. **b** Occupancy of each scFv before and after sorting using Pert-AF647 at 0.1 nM. Cell library was incubated with 0.1 nM Pert-AF647 and the

rAb(−) cell population was collected for NGS analysis. **c** Relative enrichment ratio of each scFv calculated from the occupancy in (**b**), using Niv as a reference. **d** Occupancy of each scFv before and after sorting using Pert-AF647 at 10 nM. The same cell library in (**b**) was incubated with 10 nM Pert-AF647 and the rAb(−) cell population was collected for NGS analysis. **e** Relative enrichment ratio of each scFv calculated from the occupancy in (**b**) (library cells) and (**d**), using Niv as a reference. **f** Parallel qAb classification via Epitope Binning-seq. The 14 qAbs were classified into three bins, pertuzumab-epitope, trastuzumab-epitope, and ungrouped.

mechanisms of action. By contrast, epitope-targeted drug discovery uses existing information to develop drugs with clearly defined mechanisms and therapeutic effects, thereby significantly reducing the time and risks associated with drug development. Although various direct approaches have been developed to group antibodies based on their epitopes, to the best of our knowledge, almost all of these approaches involve the production and purification of individual antibodies, which is a bottleneck in the simultaneous

evaluation of large numbers of antibodies. In this study, we have successfully developed an epitope binning platform, Epitope Binning-seq, for the parallel evaluation of numerous genetically encoded antibodies. In this platform, qAb is displayed on the cell surface of antigen-expressing cells to evaluate binding to the antigen. Subsequently, the rAb-non-binding cell population is sorted, and NGS analysis is used to identify qAbs sharing epitopes similar to those of rAb. The number of qAbs that can be evaluated using this simple and direct

platform depends on the size of the cell library, the number of cells to be sorted, and the number of NGS reads, but it is feasible that up to 10 million different qAbs could be simultaneously evaluated using mammalian display in a laboratory-level experiment. Therefore, this platform can be applied for epitope binning of both natural and artificial antibody libraries, such as polyclonal antibody pools produced by animal immunization and antibody libraries generated by random mutation.

Epitope Binning-seq segregated epitope-similar qAbs with high reliability and sensitivity. In the analysis using Pert-AF647. Notably, the negative control qAb Niv, despite an initial high abundance (36%), was not enriched after sorting (Fig. 6b, c). In the analysis using Tras-AF647, Tra-G103A, initially representing only 0.018%, was notably enriched (Supplementary Fig. 7a, b), indicating the ability to identify and evaluate one clone from a pool of at least 5500 species. Using Epitope Binning-seq, 14 qAbs were accurately classified into corresponding epitope bins without any false results observed, demonstrating the accuracy of this platform in binning analysis.

The Carterra LSA platform integrates flow microfluidics with high-throughput surface plasmon resonance, representing cutting-edge technology for characterizing antibody interactions and epitope binning. This platform significantly expedites the discovery of therapeutic antibodies, exemplified by the identification of the approved therapeutic bamlanivimab against COVID-19 within three months[35]. The Carterra LSA is able to simultaneously analyze combinations of 384 qAbs and 384 rAbs in a single run, whereas Epitope Binning-seq has the potential to assess combinations of over 10 million qAbs and multiple rAbs. Moreover, the Carterra LSA requires the production of individual qAbs, while Epitope Binning-seq eliminates this requirement, providing a significant advantage for large-scale binning applications. These characteristic differences make Epitope Binning-seq particularly suitable for analyzing a substantial number of qAbs in the early phases of drug discovery. In contrast, the Carterra LSA is most effective for examining epitope similarity among qAbs during the subsequent stages of drug discovery, particularly when the pool of candidate qAbs has been narrowed down to some extent.

Simultaneous analysis with dual rAbs highlights the potential to evaluate multiple epitopes for a larger number of qAbs using various rAbs. By combining our approach with multicolor flow cytometry, wherein rAbs are labeled with distinct fluorophores, this may enable the analysis of more epitope groups in a single experiment. Such a multiplexed epitope binning approach would substantially expand the antibody candidates pool and provide more epitope combinations for developing engineered antibody drugs, such as bispecific biologics that have two antigen-binding arms for distinct epitopes[36,37]. Furthermore, the successful enrichment of epitope-similar scFvs from the qAb library (Fig. 6b, c and Supplementary Fig. 7a, b) demonstrates the utility of Epitope Binning-seq in epitope-specific antibody screening. While various epitope-directed selection strategies exist, such as animal immunizations with epitope-containing polypeptides[38,39] or recombinant fusion proteins[40,41], the removal of an off-target binder using epitope-masked antigen mutants[42,43], and incorporating noncanonical photoreactive amino acids near the desired epitope to capture specific binders[44], these methods often require engineered antigens, which may not preserve their native conformations. In contrast, our approach enables epitope-directed antibody screening using antigen that retains its native conformation without the necessity of introducing any mutations, which is particularly advantageous for identifying conformational epitope-specific antibodies.

In our system, qAbs efficiently mask epitope on antigen when the qAb/antigen molecular ratio is high, preferring the collection of rAb(−) cell populations (Fig. 3d, e and Supplementary Fig. 3c, d). However, it is also important that the antigen is expressed above a minimal level to ensure the sensitivity of this system. If the expression level is extremely low, the binding of rAb cannot be detected, regardless of the epitope of qAb. These cells would be classified as rAb(−) cell population and are one of the reasons of false positive results. To eliminate this potential issue, it is useful to establish an antigen-expressing cell line with a detectable antigen level before introducing the gene encoding qAbs into the cells.

Interestingly, Epitope Binning-seq showed that high-affinity qAbs could be specifically selected at high rAb concentrations (Fig. 6e and Supplementary Fig. 7d). In competitive binding, the relative amount of qAb and rAb determines whether qAb-antigen or rAb-antigen interactions predominate. Under conditions of low rAb concentrations, qAbs preferentially bind to antigens on the cells because there are few competitor rAbs for the epitope. Conversely, high concentrations of rAb exert intense competitive pressure on antigen-bound qAbs to be replaced. High-affinity qAbs can withstand the pressure and keep binding to the antigen, whereas low-affinity qAbs surrender to the pressure and are replaced by the rAb. Consequently, high-affinity qAbs were selectively enriched when evaluated with a high concentration of rAb, while a low concentration of rAb allowed the identification of low-affinity qAbs as well. This notable feature, which is contingent on the rAb concentration, suggests that our system achieves epitope binning with consideration of antigen-binding affinity.

Epitope Binning-seq has the potential to be applied for antibody affinity maturation. As such, parental and mutant antibodies would be used as rAb and qAb, respectively, and Epitope Binning-seq would identify qAbs with higher binding affinity than rAb. Previously, an affinity maturation method based on competitive antigen binding between parental and mutant antibodies has been studied, in which mutant antibodies were secreted from the engineered chimera antigen-expressing mammalian cells and bound to the antigen to competitively inhibit the binding of the engineered parental antibody that caused chimera antigen dimerization induced by a small compound and then activated cell death signaling[45]. This method successfully identified affinity-matured anti-HER2 antibodies; however, it has some potential weaknesses such as a high-false positive rate caused by the binding of secreted mutant antibodies to the antigens on nearby cells and limitations on the applicable antigens on which complex engineering cannot be performed. By contrast, our platform allows mutant antibodies to be displayed on cells with little interference from other cells and can be applied to a variety of native antigens, including single-pass membrane proteins, G-protein coupled receptors, and channel proteins. In other words, Epitope Binning-seq has the potential to be a more accurate and versatile affinity maturation method than existing methods.

In conclusion, we provide proof-of-concept for our DNA sequencing-based epitope binning platform, Epitope Binning-seq. Epitope Binning-seq is a highly versatile platform for parallel antibody classification that can be applied to a wide range of antibodies and antigens. Its utility in epitope binning and potential applications in antibody identification, such as epitope-directed selection and affinity maturation, may significantly facilitate antibody drug development in the future.

## Materials and methods
### MD simulations
All MD simulations were executed using the AMBER 16 program package[46] on TSUBAME (Global Scientific Information and Computing Center at Tokyo Institute of Technology). The initial coordinates for the pertuzumab Fab-HER2 and trastuzumab Fab-HER2 complexes were obtained from PDB accession codes 1S78 and 1N8Z, respectively. The AMBER ff14SB force field was used to describe the proteins along with the TIP3P water model. The systems were fully solvated with explicit solvent and two Na+ counterions were included for electrostatic neutrality. Energy minimization and equilibration with backbone restraints were performed to optimize the systems. Subsequently, a 100 ns production run was executed, and the final 10 ns of the trajectory were used by the molecular mechanics/generalized Born surface area (MM/GBSA) module for computing the binding free energy.

### Plasmid construction
All recombinant DNA experiments were performed with the approval of the Recombinant DNA Experimental Safety Management Committee of Tokyo Institute of Technology (no. I2021017).

To express HER2, we inserted a cDNA encoding human HER2, a T2A self-cleaving peptide (EGRGSLLTCGDVEENPGP), and mCherry into the

multiple cloning site of the CSII-CMV-MCS plasmid (RIKEN Bio-Resource Center, Ibaraki, Japan) and named the resulting plasmid pCSII/HER2-mCherry.

For surface display of scFvs on cells, the DNA fragment of scFv(Dac) in the previously constructed plasmid pCSII/scFv-sfGFP[27] was replaced with that of other scFvs. The resultant plasmids expressed sfGFP together with scFv-fused proteins. To express BFP together with scFv-fused proteins, the DNA fragment of sfGFP was replaced with that of BFP, resulting in the construction of pCSII/scFv-BFP plasmids.

To secrete the scFvs extracellularly, a cDNA encoding fusion protein, which consisted of the PD-L1 secretion signal (amino acids 1–18, UniProt Q9NZQ7), scFv, single chain fragment of two human IgG1 Fc domains (amino acids 104–330, UniProt P01857), His6 tag peptide, T2A peptide, and sfGFP, was inserted into the multiple cloning site of CSII-CMV-MCS plasmid. The resulting plasmid was named pCSII/scFv-scFc-H6.

## Cell culture
The human chronic myelogenous leukemia cell line K562 and subclones of the human embryonic kidney cell lines HEK293, HEK293T, and the Lenti-X 293 T cell line were obtained from JCRB Cell Bank (Osaka, Japan), RIKEN Bio-Resource Center, and Clontech (Mountain View, CA, USA), respectively. The K562/HER2, K562/HER2/scFv, and K562/HER2/scFv^B cell lines were established through lentiviral transduction of K562 and K562/HER2 cells with the plasmids pCSII/HER2-mCherry, pCSII/scFv-sfGFP, and pCSII/scFv-BFP, respectively.

K562, K562/HER2, K562/HER2/scFv, and K562/HER2/scFv^B cells, as well as HEK293T and Lenti-X 293 T cells, were maintained in 10% FBS-RPMI-1640 (Invitrogen, Waltham, MA, USA) and 10% FBS-DMEM (Nacalai Tesque, Kyoto, Japan), respectively. All media were supplemented with penicillin (100 U/mL) and streptomycin (100 μg/mL) (Nacalai Tesque). Cells were maintained in a 37 °C incubator with 5% $CO_2$ and regularly checked for mycoplasma contamination using a mycoplasma test kit (Lonza, Basel, Switzerland).

## Lentivirus transduction
Plasmid pCSII/HER2-mCherry, pCSII/scFv-sfGFP, pCSII/scFv-BFP, or pCSII/scFv-scFc-H6 was mixed with Lentiviral High Titer Packaging Mix (Takara Bio, Shiga, Japan). Lenti-X 293 T cells were then transfected with the plasmid mixture using Lipofectamine LTX (Invitrogen). After incubation for 24 h, the medium was replaced with 10% FBS-DMEM containing 10 μM forskolin (Fujifilm Wako Pure Chemical, Osaka, Japan). At 48 h after transfection, the lentiviruses in the medium were collected and concentrated using a Lenti-X Concentrator (Clontech) in accordance with the manufacturer's instructions.

To infect K562 and K562/HER2 cells with lentiviruses carrying genes from pCSII/HER2-mCherry, pCSII/scFv-sfGFP, or pCSII/scFv-BFP, we employed a modified spinoculation protocol[47,48]. The cells were transduced at a multiplicity of infection of 0.3 to maximize the cell population with a single copy of the target gene in the genome. First, the cells were mixed in medium with the lentivirus and 8 μg/mL polybrene (Nacalai Tesque). The cell suspension was then centrifuged at $800 \times g$ for 90 min at room temperature. The resulting cell pellet was resuspended in fresh medium and cultured. The transduction efficiency was confirmed by cell observation using a confocal microscope Zeiss LSM780 (Zeiss, Jena, Germany) with the appropriate filters (Ex/Em = $545 \pm 25$ nm/$605 \pm 70$ nm for mCherry, Ex/Em = $470 \pm 40$ nm/$520 \pm 50$ nm for sfGFP, and Ex/Em = 365 nm/$445 \pm 50$ nm for BFP).

To infect HEK293T cells with lentivirus carrying genes from pCSII/scFv-scFc-H6, the lentivirus was resuspended in medium with 8 μg/mL polybrene. The mixture was then added to approximately 80%–90% confluent HEK293T cells. After incubation for 48 h, the transduction efficiency was verified under a Biorevo BZ-710 fluorescence microscope (Keyence, Osaka, Japan) with the appropriate filters (Ex/Em = $470 \pm 40$ nm/$520 \pm 50$ nm).

## FCM analyses and cell sorting
To confirm the cell surface localization of HER2 and scFv-fused proteins, $2 \times 10^5$ K562, K562/HER2, and K562/HER2/scFv cells were stained with AF488-conjugated anti-human HER2 antibody (clone 24D2, 1:100; BioLegend, San Diego, CA, USA) and AF647-conjugated anti-His tag antibody (0.5 μg/mL; MBL Life Science, Tokyo, Japan), respectively, for 1 h at 4 °C prior to FCM analysis.

To evaluate epitope similarity, $2 \times 10^5$ K562/HER2/scFv, and K562/HER2/scFv^B cells were stained with 0.1, 1, 10, and 100 nM Pert-AF647, Tras-AF647, and Tras-AF488, which were prepared by conjugating fluorescent dyes AF647 and AF488 to pertuzumab (MedChemExpress, Monmouth Junction, NJ, USA) and trastuzumab (Chugai pharmaceutical, Tokyo, Japan) using the Alexa Fluor 647 or Alexa Fluor 488 Antibody Labeling Kit (Invitrogen), respectively, for 1 h at 4 °C prior to FCM analysis.

To analyze the HER2-binding affinity of scFvs, $2 \times 10^5$ K562/HER2 cells were incubated with 0.5, 1, 5, 10, 25, 50, 100, 200, 400, 800, and 1600 nM scFv-scFc-H6 proteins for 90 min at 4°C, then stained with AF488-conjugated anti-HisTag antibody (clone J099B12, 1:400; BioLegend) for 40 min at 4 °C prior to FCM analysis.

To collectively evaluate epitope similarity, $1 \times 10^6$ library cells consisting of 14 different K562/HER2/scFv cells were stained with 0.1 and 10 nM Pert-AF647 and Tras-AF647 for 1 h at 4°C prior to cell sorting.

FCM data were acquired using flow cytometers, including an iCyt ec800 flow cytometer (Sony Biotechnology, Tokyo, Japan), a FACSCanto II (BD Biosciences, Franklin Lakes, NJ, USA), or a FACSAria III (BD Biosciences). The Pert-AF647- and Tras-AF647-non-binding cell populations were sorted with a FACSAria III. The data from FCM and cell sorting were analyzed using FlowJo software ver.10.8.1 (FlowJo, LLC, Ashland, OR, USA). In all experiments, 3% FBS-PBS was used for protein and antibody dilution and cell washing.

## Protein purification
HEK293T cells secreting the scFv-scFc-H6 proteins were cultured in 1% FBS-DMEM medium for 48 h and then the culture supernatant was collected. The proteins were purified from the supernatant using Ni-NTA agarose (QIAGEN, Hilden, Germany) according to the manufacturer's guidelines. The purity of the purified protein was confirmed by silver staining using Silver Stain KANTO III (KANTO CHEMICAL, Tokyo, Japan).

## NGS analyses
Genomic DNA from the library and from sorted cells was extracted using a GenElute Mammalian Genomic DNA Miniprep Kit (Sigma-Aldrich, St. Louis, MO, USA) according to the manufacturer's instructions. The full-length scFv gene was amplified from the genomic DNA using a forward PCR primer annealing to the signal sequence (5'-CGGTCTCGA-GATGAGGATATTTGCTGTCTTTATATTCATG-3') and a reverse PCR primer annealing to the linker sequence (5'- CTTCCTCGCTAAT-CAGTTTCTGTTCTC-3'), which flank the N and C-termini of scFv, respectively. DNA fragments containing the sequences of the mutated region were amplified by a second nested PCR using a template from the first PCR, with a forward primer with a unique 6-bp barcode (5'-barcode-TGGGTAAGACAAGCTCCAGG-3') and a reverse primer (5'-CCTTG TCCCCAATAGTC-3'). All the PCR amplifications were carried out using a KOD FX kit (TOYOBO, Osaka, Japan) and the aforementioned primers (Eurofins Genomics, Tokyo, Japan). The PCR products were purified using AMPure XP magnetic beads (Beckman Coulter Life Sciences, Indianapolis, IN, USA) according to the manufacturer's instructions. Targeted NGS was carried out on the NovaSeq 6000 system by Macrogen Japan Corp. (Tokyo, Japan) with a read depth of 2,600,000 reads.

To filter out the low-quality reads from the raw data, Trimmomatic (version 0.39)[49] was used with the MINLEN parameter set to 50 and SLIDINGWINDOW set to 20:20. To ensure high sequencing accuracy, only reads that matched the primer sequences were retained for downstream

analysis. Then, reads were assigned to different samples according to unique barcodes. These reads were then translated into amino acid sequences using Biostrings (version 2.66.0-1)[50] and assigned to corresponding antibody clones. The occupancy of each antibody in the samples was calculated by dividing the count of qAb by the total count of a given sample. The enrichment ratio was calculated by comparing the occupancy of qAb in sorted cells to those in the original library. Then, the enrichment ratio value of the negative control Niv was set to 1 and the relative enrichment ratio of other qAbs compared to Niv was calculated. When the relative enrichment ratio was greater than 3, it was considered to be substantially enriched.

## Statistics and reproducibility

All experimental data were collected from three independent experiments and are presented as the mean ± SEM unless otherwise stated. GraphPad Prism 10 (GraphPad Software, Boston, MA, USA) was used for statistical analysis. $P$ values were calculated using Tukey's multiple comparisons test or two-side unpaired Student's $t$ tests as described in the figure legends, and $P < 0.05$ was considered statistically significant.

## Reporting summary

Further information on research design is available in the Nature Portfolio Reporting Summary linked to this article.

## Data availability

The source data underlying Figs. 2d, e, 3a, b, d, e, 5, 6b–e, Supplementary Figs. 1, 3, 4b, 5, 7 are provided as Supplementary Data 1. Raw FASTQ files generated from DNA sequencing have been deposited in the DNA Data Bank of Japan (DDBJ) under the accession numbers of BioProject: PRJDB18047 and Run: DRR550467-DRR550468. All other data generated during this study are available from the corresponding author upon reasonable request.

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

## Acknowledgements
We thank the Biomaterial Analysis Division of the Open Facility Center at Tokyo Institute of Technology for performing DNA sequencing, as well as the FACS Core Laboratory at the Institute of Medical Science, University of Tokyo for aiding the operation of the FACS Aria III and the FACSCanto II. This study was supported by AMED under grant numbers JP19am0401023h0001, JP20am0401023h0002, JP21am0401023h0003, JP22am0401023h0004, and JP23am0401023h0005 (to T.K.). N.L. was supported by the China Scholarship Council. We thank Edanz (https://jp.edanz.com/ac) for editing a draft of this manuscript.

## Author contributions
T.K. and S.K.-K. conceived and supervised the whole project. N.L. performed the experiments and collected and analyzed data. K.M. contributed to the next-generation sequencing raw data processing. T.O. contributed to the MD analysis, and S.S. supported the cell sorting experiments. N.L., S.K.-K., and T.K. wrote the manuscript.

## Competing interests
The authors declare no competing interests.
