## [Peer review file · Communications Biology]

Reviewers' comments:

Reviewer #2 (Remarks to the Author):

In this article, the authors present a novel technique to profile antibody binding locations on antigen using sequencing, and thus, potentially at high throughput. The present paper represents a proof of principle towards that. The authors use carefully selected controls to benchmark and test their approach. The experiments are carefully designed, and the text reads well. I have only minor comments.

- title: I suggest rewording the title and rethinking the use of the word "grouping". Grouping by what? It's a bit unclear (I know it's by epitope but there are more elegant solutions of wording this).
- how sensitive is the authors' approach to partial epitope overlap? In other words, have the authors measured how similar two epitopes need to be to be considered the same?
- can the authors comment on how this tool compares to carterra?

Reviewer #3 (Remarks to the Author):

This manuscript reports a DNA sequencing-based epitope binning approach, termed Epitope Binning-seq, to group multiple monoclonal antibodies simultaneously. The authors displayed the query antibodies (qAbs) on antigen-expressing cells, and the resulting cells treated with a fluorescently-labeled reference antibody (rAb) targeting a desired epitope followed by analysis using flow cytometry. The authors employed the human epidermal growth factor receptor 2 (HER2) and HER2-targeting monoclonal antibodies, pertuzumab and trastuzumab, as model antigen and antibodies, respectively. The authors showed that the qAbs with epitope similar to the rAb can be identified by next-generation sequencing analysis of fluorescence-negative cells. In addition, sensitivity and reliability of the system were verified using rAbs. The developed system was revealed to allow simultaneous epitope evaluation of 14 qAbs at various abundances in libraries, grouping them into respective epitope bins. This manuscript provides efficient epitope-binding approach for identifying antibodies with new functionality at the early discovery stage. I suggest publication in the journal after revision.

1. Basically, the epitope-binning system is based on the competitive binding assay between query antibodies (qAbs) and reference antibody (rAb) for an existing antigen. In this regard, the title of the manuscript seems to be misleading, and should be changed to reflect the underlying principle.
2. To more precisely group antibodies according to their epitopes, more reference antibodies should be used. In this study, HER2-targeting two monoclonal antibodies, pertuzumab and trastuzumab, were used as reference antibodies, since only limited number of HER2-targeting monoclonal antibodies are currently available. Thus, the practical utility of the developed system would be mainly limited by the number of reference Abs with different epitopes. Are there any ways of overcoming this situation?
3. Epitope-binning of qAbs was mainly achieved by analyzing the portion of rAb-non-binding cells using flow cytometry. How about the false-positive rate of the binning system?
4. Binding affinity of ScFv for an antigen is known to be quite different from the whole Ab. This might affect the efficiency of the competitive binding assay. Did the authors check the binding affinity of ScFv compared to the whole antibody?

Point-by-point response to reviewers' comments

Reviewer #2 (Remarks to the Author)

In this article, the authors present a novel technique to profile antibody binding locations on antigen using sequencing, and thus, potentially at high throughput. The present paper represents a proof of principle towards that. The authors use carefully selected controls to benchmark and test their approach. The experiments are carefully designed, and the text reads well. I have only minor comments.

Thank you for your valuable comments and suggestions that helped improve our manuscript. The revised words and sentences are colored in blue in the revised manuscript.

1. title: I suggest rewording the title and rethinking the use of the word "grouping". Grouping by what? It's a bit unclear (I know it's by epitope but there are more elegant solutions of wording this).

Answer: Thank you for your suggestion. We have changed the title and relating sentences to describe the characteristics of our system more clearly in the revised manuscript (lines 1-2, 23, and 90-91, and 93).

(lines 1-2) Epitope binning for multiple antibodies simultaneously using mammalian cell display and DNA sequencing

(line 23) Here, we established Epitope Binning-seq, an epitope binning platform for simultaneously analyzing multiple antibodies.

(lines 90-91) In this way, we developed a parallel epitope binning platform for multiple qAbs named Epitope Binning-seq.

(line 93) Epitope Binning-seq takes advantage of genetically encoded qAbs displayed on the surface of antigen-expressing cells to enable the simultaneous evaluation of epitope similarity for a large number of qAbs by next-generation sequencing (NGS) without the need for individual antibody purification.

2. How sensitive is the authors' approach to partial epitope overlap? In other words, have the authors measured how similar two epitopes need to be to be considered the same?

Answer: Thank you for the comment. Determining the degree of overlap between two epitopes requires detailed information about the antibody-binding residues on the antigen. This information can only be obtained by high-resolution analysis using epitope mapping technologies and cannot be revealed by epitope binning techniques, including Epitope Binning-seq. Therefore, we have not quantified the level of similarity at which two epitopes are considered the same, and it is difficult to show results that can answer the reviewer's questions in this study. However, we believe that the reviewer question is important in analyzing epitopes, and in future research, we would like to develop a system that can address this question.

3. Can the authors comment on how this tool compares to carterra?

Answer: We thank your valuable suggestion. The Carterra LSA is cutting-edge technology for characterizing antibody interactions and epitope binning. We have compared the characteristics between Epitope Binning-seq and the Carterra LSA, and added text in the Discussion section and reference to the Carterra LSA in the revised manuscript (lines 330-344, lines 647-648).

(lines 330-344) The Carterra LSA platform integrates flow microfluidics with high-throughput SPR, representing cutting-edge technology for characterizing antibody interactions and epitope binning. This platform significantly expedites the discovery of therapeutic antibodies, exemplified by the identification of the approved therapeutic bamlanivimab against COVID-19 within three months³⁵. The Carterra LSA is able to simultaneously analyze combinations of 384 qAbs and 384 rAbs in a single run, whereas Epitope Binning-seq has the potential to assess combinations of over 10 million qAbs and multiple rAbs. Moreover, the Carterra LSA requires the production of individual qAbs, while Epitope Binning-seq eliminates this requirement, providing a significant advantage for large-scale binning applications. These characteristic differences make Epitope Binning-seq particularly suitable for analyzing a substantial number of qAbs in the early phases of drug discovery. In contrast, the Carterra LSA is most effective for examining epitope similarity among qAbs during the subsequent stages of drug discovery, particularly when the pool of candidate qAbs has been narrowed down to some extent.

(lines 647-648) 35. Jones, B. E. et al. The neutralizing antibody, LY-CoV555, protects against SARS-CoV-2 infection in nonhuman primates. *Sci Transl Med* **13** (2021).

Reviewer #3 (Remarks to the Author)

This manuscript reports a DNA sequencing-based epitope binning approach, termed Epitope Binning-seq, to group multiple monoclonal antibodies simultaneously. The authors displayed the query antibodies (qAbs) on antigen-expressing cells, and the resulting cells treated with a fluorescently-labeled reference antibody (rAb) targeting a desired epitope followed by analysis using flow cytometry. The authors employed the human epidermal growth factor receptor 2 (HER2)

and HER2-targeting monoclonal antibodies, pertuzumab and trastuzumab, as model antigen and antibodies, respectively. The authors showed that the qAbs with epitope similar to the rAb can be identified by next-generation sequencing analysis of fluorescence-negative cells. In addition, sensitivity and reliability of the system were verified using rAbs. The developed system was revealed to allow simultaneous epitope evaluation of 14 qAbs at various abundances in libraries, grouping them into respective epitope bins. This manuscript provides efficient epitope-binding approach for identifying antibodies with new functionality at the early discovery stage. I suggest publication in the journal after revision.

Thank you for your valuable comments and suggestions that helped improve our manuscript. The revised words and sentences are colored in blue within the revised manuscript.

1. Basically, the epitope-binning system is based on the competitive binding assay between query antibodies (qAbs) and reference antibody (rAb) for an existing antigen. In this regard, the title of the manuscript seems to be misleading, and should be changed to reflect the underlying principle.

Answer: Thank you for your suggestion. We have changed the title and relating sentences to describe the characteristics of our system more clearly in the revised manuscript (lines 1-2, 23, and 90-91, and 93).

(lines 1-2) Epitope binning for multiple antibodies simultaneously using mammalian cell display and DNA sequencing

(line 23) Here, we established Epitope Binning-seq, an epitope binning platform for simultaneously analyzing multiple antibodies.

(lines 90-91) In this way, we developed a parallel epitope binning platform for multiple qAbs named Epitope Binning-seq.

(line 93) Epitope Binning-seq takes advantage of genetically encoded qAbs displayed on the surface of antigen-expressing cells to enable the simultaneous evaluation of epitope similarity for a large number of qAbs by next-generation sequencing (NGS) without the need for individual antibody purification.

2. To more precisely group antibodies according to their epitopes, more reference antibodies should be used. In this study, HER2-targeting two monoclonal antibodies, pertuzumab and trastuzumab, were used as reference antibodies, since only limited number of HER2-targeting monoclonal antibodies are currently available. Thus, the practical utility of the developed system

would be mainly limited by the number of reference Abs with different epitopes. Are there any ways of overcoming this situation?

Thank you for asking about an important issue. Increasing the number of rAbs is helpful for maximizing the epitope coverage and facilitating high-resolution binning analysis in the practical use of our system. When well-studied antibodies against a particular antigen are limited or unknown, using qAbs as rAbs can provide an alternative option. It is common to utilize qAbs to be analyzed as rAbs, as seen in Fig. A [*MAbs*, 9(1), 29–42 (2017)], with each antibody serving as both the ligand (rAb) and analyte (qAb) in pairwise competitive binding assays. This strategy used in conventional epitope binning methods can also be applied to our Epitope Binning-seq. Some or all of qAbs can be selected and prepared as rAbs for the analysis.

Fig. A. Heat map summarizing the epitope binning results of seven antibodies. Each analyte (qAb) is tested for their binding to the antigen captured by the ligand (rAb) using biolayer interferometry. Green and red cells represent non-blocking and blocking pairs of antibodies, respectively. Cited from *MAbs*, 9(1), 29–42 (2017).

3. Epitope-binning of qAbs was mainly achieved by analyzing the portion of rAb-non-binding cells using flow cytometry. How about the false-positive rate of the binning system?

Thank you very much for the valuable comment that indicate points for improving our method. In the original manuscript, we utilized the enrichment ratio of 1 as the cutoff to differentiate whether a qAb is enriched or not. Some epitope-dissimilar qAbs showed an enrichment ratio slightly exceeding 1 (highlighted in yellow in Table 1), resulting in false positive rates of 21.4% (3/14 qAbs) and 14.3% (2/14 qAbs), respectively. As discussed in the original manuscript, this may be attributed to the broad setting of the gating region during cell sorting, which may introduce

some undesired cells. This can be addressed by narrowing the sorting region or optimizing other parameters of the cell sorter to ensure entire separation of rAb (-) and rAb (+) populations. Therefore, we have reconsidered the analytical parameters. As a result, we were able to set a relative enrichment ratio of 3 as the cutoff for identifying significantly enriched qAbs. This change allowed 14 qAbs to be accurately classified into their corresponding epitope bins without any false results, demonstrating the increased accuracy of this platform in the binning analysis. We have revised Fig. 6c, 6e and 6f, Supplementary Fig. 6b and 6d, and relating sentences to explain the analysis in the revised manuscript (lines 277-290, 292, 294, 327-329 and 539-542)

(lines 277-290) Upon collecting the rAb(-) cell population that reacted with 0.1 nM Pert-AF647, the percentage of cells displaying Per and its mutants increased significantly (Fig. 6b), with all showing a relative enrichment ratio greater than 3 (Fig. 6c), indicating the significant enrichment of desired qAbs through our epitope binning approach. In contrast, the relative enrichment ratios of Tra and its mutants (which have different epitopes from pertuzumab), and Dac and Niv (which bind to different antigens than HER2), were less than 3. Furthermore, when the reaction concentration of Pert-AF647 was increased to 10 nM, only Per and Per-P53A, which have strong binding affinity, showed a relative enrichment ratio above 3 (Fig. 6d, 6e). When the rAb was changed to 0.1 nM of Tras-AF647, Tra and its mutants exhibited substantial enrichment, all showing the relative enrichment ratio greater than 3, much greater than those of other qAbs (Supplementary Fig. 6a, 6b). As expected, the slight enrichment of scFvs with relatively high binding affinity was observed at the rAb concentrations of 10 nM (Supplementary Fig. 6c, 6d). In this study case, through the identification of significantly enriched antibody clones under the guidance of each rAb at the concentration of 0.1 nM, the 14 qAbs were grouped into three distinct epitope bins: pertuzumab, trastuzumab, and ungrouped (Fig. 6f).

(lines 292 and 294) Specifically, Per and Per mutants, along with Tra and Tra mutants, were classified into the respective pertuzumab- and trastuzumab-epitope groups, which was consistent with their origins, while qAbs Dac and Niv, which exhibited no binding to the antigen, did not belong to either of the two rAb bins.

(lines 327-329) Using Epitope Binning-seq, 14 qAbs were accurately classified into corresponding epitope bins without any false results observed, demonstrating the accuracy of this platform in the binning analysis.

[This sentence has been replaced from the following original discussion: It is worth noting that Per and Niv exhibited an enrichment ratio slightly exceeding 1 (Supplementary Fig. 6d), possibly resulting from the broad setting of the gating region during cell sorting. This can be addressed by narrowing the sorting region or optimizing other parameters of the cell sorter to ensure entire

separation of rAb (-) and rAb (+) populations. Another factor to consider was the high initial abundance of Per and Niv, accounting for 52% and 26% of qAbs, respectively (Supplementary Fig. 6a), implying that an even abundance of each qAb in the initial library may enhance epitope binning accuracy.]

(lines 539-542) Then, the enrichment ratio value of the negative control Niv was set to 1 and the relative enrichment ratio of other qAbs compared to Niv was calculated. When the relative enrichment ratio was greater than 3, it was considered to be significantly enriched.

Table 1. Enrichment ratios of qAbs in the analysis of AF647-Tras

qAbs	0.1 nM	10 nM
Per	0.998	1.106
Per-N52A	1.053	0.436
Per-P53A	0.678	0.076
Per-N54A	1.202	0.039
Per-S55A	1.202	0.068
Per-G56A	0.967	0.114
Tra	5.181	2.913
Tra-G101A	3.489	1.369
Tra-D102A	8.684	2.866
Tra-G103A	6.775	2.371
Tra-F104A	3.666	0.753
Tra-Y105A	5.672	0.484
Dac	0.797	0.817
Niv	0.966	1.428

4. Binding affinity of ScFv for an antigen is known to be quite different from the whole Ab. This might affect the efficiency of the competitive binding assay. Did the authors check the binding affinity of ScFv compared to the whole antibody?

We thank your important comment. We have measured the binding affinity of whole Abs, pertuzumab and trastuzumab using K562/HER2 cells. The K_D values of pertuzumab and trastuzumab were 0.29 nM and 1.3 nM, respectively (Fig. B). These values were several times lower than their scFvs (Per=9.0 nM and Tra=13 nM) (Fig. 5a). We believe this difference is probably due to the conversion of binding mode from bivalent to monovalent, as seen in previous studies [*PLoS ONE*, 10(4), e0124440 (2015), *Mol Cancer Ther*, 11 (7), 1467–1476 (2012)].

Fig. B. Binding of whole Abs to HER2

K562/HER2 cells were incubated with the Pert-AF647 and Tras-AF647 at concentrations of 0.005, 0.01, 0.05, 0.1, 0.5, 1, 5, 10, and 25 nM for 90 min at 4°C, followed by FCM analysis.

The binding affinity of rAb directly impacts the sensitivity of the competitive binding assay because the competitive pressure is influenced by both the binding affinity and concentration of the rAb. In our system, we have verified that decreasing the concentration of rAb leads to a reduction in competitive pressure and an increase in the detection sensitivity for mutant scFvs with lower affinity (Fig. 5b and 5c, Supplementary Fig. 5a and 5b). If the efficiency of the competitive binding assay is compromised due to a substantial difference in binding affinity between the whole rAb and the scFv qAb, this efficiency can be enhanced by reducing the concentration of the rAb.

Additional revisions

We have made additional revisions as follows:

(lines 32) This versatile platform is applicable to diverse antibodies and antigens, potentially expediting the identification of clinically useful antibodies.

(line 69) Various competitive immunoassay formats (such as classical sandwich, premix, or in-tandem assays) can be used in conjunction with an enzyme-linked immunosorbent assay²⁰, biolayer interferometry, or surface plasmon resonance (SPR)²¹⁻²⁴.

(line 97) We then achieved epitope similarity assessment of four qAb-rAb pairs using dual rAbs with distinct epitopes simultaneously.

(line 214) As a result of the similar emission/excitation wavelengths of AF488 and sfGFP, the qAb cDNA was reconstructed to co-express with blue fluorescent protein (BFP) (Fig. 4a).

(line 224) Furthermore, when the cells displaying different qAbs were mixed pairwise at a 1:1 ratio and evaluated using two rAbs with different epitope recognition, the presence of cells expressing scFv(Per) or scFv(Tra) sharing the same epitope with either Pert-AF647 or Tras-AF488 led to the observation of corresponding populations that exhibited no binding to Pert-AF647 or Tras-AF488 (Fig. 4d).

(line 228) These results suggested that using two distinct rAbs enables the simultaneous evaluation of four rAb-qAb pairs with distinct epitopes in a single experiment.

(line 243) Wild-type and mutant scFvs, scFv(Per), N52A mutant scFv(Per), P53A mutant scFv(Per), N54A mutant scFv(Per), S55A mutant scFv(Per), G56A mutant scFv(Per), scFv(Tra), G101A mutant scFv(Tra), D102A mutant scFv(Tra), G103A mutant scFv(Tra), F104A mutant scFv(Tra), Y105A mutant scFv(Tra), scFv(Dac), and scFv of PD-1 targeting nivolumab [scFv(Niv)] (Supplementary Fig. 2a) are hereafter referred to as Per, Per-N52A, Per-P53A, Per-N54A, Per-S55A, Per-G56A, Tra, Tra-G101A, Tra-D102A, Tra-G103A, Tra-F104A, Tra-Y105A, Dac, and Niv, respectively.

(line 326) In the analysis using Tras-AF647, Tra-G103A, initially representing only 0.018%, was notably enriched (Supplementary Fig. 6a, 6b), indicating the ability to identify and evaluate one clone from a pool of at least 5500 species.

(lines 714, 719, 728, 735, 745, 750, 771, and 776) Sample numbers were added in the corresponding figure legend.

Updating figures

(1) We have updated Figure 6c, 6e, 6f, and Supplementary Figure 6b and 6d.

Figure 6. Epitope Binning-seq for analyzing various qAbs

Supplementary Figure 6. Epitope Binning-seq for various qAbs using Tras-AF647

(2) We have included individual data points in the plot graphs of the following figures: Figure 2d, 2e, 3a, 3b, 3d, 3e, 5b, and 5c, as well as Supplementary figure 1, 3, 4b and 5.

Figure 2. Validation of the evaluation system using HER2-binding antibodies

Figure 3. Effects of the Pert-AF647 concentration and the balance between qAb and HER2 expression levels on evaluation sensitivity

Figure 5. Evaluation of qAbs with various HER2-binding affinities

Supplementary Figure 1. Identification of antibody-binding residues on HER2 and HER2-binding residues on antibodies

Supplementary Figure 3. Effects of the Tras-AF647 concentration and the balance between qAb and HER2 expression levels on evaluation sensitivity

b
Supplementary Figure 4. HER2-binding affinity of scFvs

a**b**
Supplementary Figure 5. Evaluation of Tra mutants with various HER2-binding affinities

Reviewers' comments:

Reviewer #4 (Remarks to the Author):

Lin et al. developed Epitope Binning-seq to identify antibodies with similar epitope binding through a competitive binding strategy, employing flow cytometry and DNA sequencing. Specifically, single-chain variable fragments (scFvs), derived from gene transduction, are expressed on the surface of cells that present the antigen. Incubation with fluorescently labeled "reference" antibodies allows for the identification of antigen-presenting cells (APCs) that exhibit no fluorescence. These APCs are presumed to bind to the same epitope as the reference antibodies because the expressed scFv competes for and blocks binding to the antigen. The system was tested using HER2-binding antibodies to evaluate the effects of the antigen and scFv concentrations on sensitivity. Additionally, the ability to identify mutated scFvs with epitope binding similar to that of reference antibodies (rAbs) at various concentrations was assessed. Finally, Epitope Binning-seq was applied to a library of 14 scFvs to determine their epitope binding similarities to the rAb.

The manuscript is well written. I have a few comments below:

1. In Figure 5b and 5c, as well as the corresponding text from lines 252 to 262, it is unclear why the affinities of those specific mutants are compared statistically. Shouldn't the comparison ideally be made with Per? The text mentions: "When cells displaying Per and its mutants were reacted with 0.1 nM Pert-AF647, the percentage of the rAb(-) cell population remained relatively consistent for most scFvs, except for a significant reduction for Per-N52A, which exhibits the lowest binding affinity (Fig. 5a, 5b)." It should be clarified against which scFvs the cell populations are being compared to remain relatively constant.
2. In Figure 3a, the fraction of Pert-AF647-non-binding cells decreases only when the concentration of Pert-AF647 increases to 100 nM, showing no change between 0.1 nM and 10 nM. This contrasts with Figure 5, where at 10 nM, the fraction of Pert-AF647-non-binding cells decreases significantly compared to 0.1 nM, presumably due to lower affinity antibodies. The authors conclude that "These results clearly indicate that the binding affinity of qAbs affects the sensitivity of epitope evaluation, and that lower rAb concentrations allow for the evaluation of both low-affinity and high-affinity qAbs." However, could this also be attributed to the binding of completely different epitopes? Therefore, it raises the question of how similar these Abs are in terms of their epitope binding. Can the epitopes of the different scFv mutations in Figure 5a be predicted using molecular dynamics in a similar manner as the epitopes identified in Figure 2b?
3. Can the authors expand on Figure 6f? Which concentration of Pert-AF647 was used to categorize the scFvs? It appears to have been 0.1 nM. However, at 10 nM, it seems the scFvs would not be grouped correctly. Therefore, would the authors conclude that at 10 nM, Epitope Binning-seq would not group antibodies with similar epitope binding accurately?

Point-by-point response to reviewers' comments

Reviewer #2 (Remarks to the Author)

In this article, the authors present a novel technique to profile antibody binding locations on antigen using sequencing, and thus, potentially at high throughput. The present paper represents a proof of principle towards that. The authors use carefully selected controls to benchmark and test their approach. The experiments are carefully designed, and the text reads well. I have only minor comments.

Thank you for your valuable comments and suggestions that helped improve our manuscript. The revised words and sentences are colored in blue in the revised manuscript.

1. title: I suggest rewording the title and rethinking the use of the word "grouping". Grouping by what? It's a bit unclear (I know it's by epitope but there are more elegant solutions of wording this).

Answer: Thank you for your suggestion. We have changed the title and relating sentences to describe the characteristics of our system more clearly in the revised manuscript (lines 1-2, 23, and 90-91, and 93).

(lines 1-2) Epitope binning for multiple antibodies simultaneously using mammalian cell display and DNA sequencing

(line 23) Here, we established Epitope Binning-seq, an epitope binning platform for simultaneously analyzing multiple antibodies.

(lines 90-91) In this way, we developed a parallel epitope binning platform for multiple qAbs named Epitope Binning-seq.

(line 93) Epitope Binning-seq takes advantage of genetically encoded qAbs displayed on the surface of antigen-expressing cells to enable the simultaneous evaluation of epitope similarity for a large number of qAbs by next-generation sequencing (NGS) without the need for individual antibody purification.

2. How sensitive is the authors' approach to partial epitope overlap? In other words, have the authors measured how similar two epitopes need to be to be considered the same?

Answer: Thank you for the comment. Determining the degree of overlap between two epitopes requires detailed information about the antibody-binding residues on the antigen. This information can only be obtained by high-resolution analysis using epitope mapping technologies and cannot be revealed by epitope binning techniques, including Epitope Binning-seq. Therefore, we have not quantified the level of similarity at which two epitopes are considered the same, and it is difficult to show results that can answer the reviewer's questions in this study. However, we believe that the reviewer question is important in analyzing epitopes, and in future research, we would like to develop a system that can address this question.

3. Can the authors comment on how this tool compares to carterra?

Answer: We thank your valuable suggestion. The Carterra LSA is cutting-edge technology for characterizing antibody interactions and epitope binning. We have compared the characteristics between Epitope Binning-seq and the Carterra LSA, and added text in the Discussion section and reference to the Carterra LSA in the revised manuscript (lines 332-346, lines 649-650).

(lines 332-346) The Carterra LSA platform integrates flow microfluidics with high-throughput SPR, representing cutting-edge technology for characterizing antibody interactions and epitope binning. This platform significantly expedites the discovery of therapeutic antibodies, exemplified by the identification of the approved therapeutic bamlanivimab against COVID-19 within three months³⁵. The Carterra LSA is able to simultaneously analyze combinations of 384 qAbs and 384 rAbs in a single run, whereas Epitope Binning-seq has the potential to assess combinations of over 10 million qAbs and multiple rAbs. Moreover, the Carterra LSA requires the production of individual qAbs, while Epitope Binning-seq eliminates this requirement, providing a significant advantage for large-scale binning applications. These characteristic differences make Epitope Binning-seq particularly suitable for analyzing a substantial number of qAbs in the early phases of drug discovery. In contrast, the Carterra LSA is most effective for examining epitope similarity among qAbs during the subsequent stages of drug discovery, particularly when the pool of candidate qAbs has been narrowed down to some extent.

(lines 649-650) 35. Jones, B. E. et al. The neutralizing antibody, LY-CoV555, protects against SARS-CoV-2 infection in nonhuman primates. *Sci Transl Med* **13** (2021).

Reviewer #3 (Remarks to the Author)

This manuscript reports a DNA sequencing-based epitope binning approach, termed Epitope Binning-seq, to group multiple monoclonal antibodies simultaneously. The authors displayed the query antibodies (qAbs) on antigen-expressing cells, and the resulting cells treated with a fluorescently-labeled reference antibody (rAb) targeting a desired epitope followed by analysis using flow cytometry. The authors employed the human epidermal growth factor receptor 2 (HER2)

and HER2-targeting monoclonal antibodies, pertuzumab and trastuzumab, as model antigen and antibodies, respectively. The authors showed that the qAbs with epitope similar to the rAb can be identified by next-generation sequencing analysis of fluorescence-negative cells. In addition, sensitivity and reliability of the system were verified using rAbs. The developed system was revealed to allow simultaneous epitope evaluation of 14 qAbs at various abundances in libraries, grouping them into respective epitope bins. This manuscript provides efficient epitope-binding approach for identifying antibodies with new functionality at the early discovery stage. I suggest publication in the journal after revision.

Thank you for your valuable comments and suggestions that helped improve our manuscript. The revised words and sentences are colored in blue within the revised manuscript.

1. Basically, the epitope-binning system is based on the competitive binding assay between query antibodies (qAbs) and reference antibody (rAb) for an existing antigen. In this regard, the title of the manuscript seems to be misleading, and should be changed to reflect the underlying principle.

Answer: Thank you for your suggestion. We have changed the title and relating sentences to describe the characteristics of our system more clearly in the revised manuscript (lines 1-2, 23, and 90-91, and 93).

(lines 1-2) Epitope binning for multiple antibodies simultaneously using mammalian cell display and DNA sequencing

(line 23) Here, we established Epitope Binning-seq, an epitope binning platform for simultaneously analyzing multiple antibodies.

(lines 90-91) In this way, we developed a parallel epitope binning platform for multiple qAbs named Epitope Binning-seq.

(line 93) Epitope Binning-seq takes advantage of genetically encoded qAbs displayed on the surface of antigen-expressing cells to enable the simultaneous evaluation of epitope similarity for a large number of qAbs by next-generation sequencing (NGS) without the need for individual antibody purification.

2. To more precisely group antibodies according to their epitopes, more reference antibodies should be used. In this study, HER2-targeting two monoclonal antibodies, pertuzumab and trastuzumab, were used as reference antibodies, since only limited number of HER2-targeting monoclonal antibodies are currently available. Thus, the practical utility of the developed system

would be mainly limited by the number of reference Abs with different epitopes. Are there any ways of overcoming this situation?

Answer: Thank you for asking about an important issue. Increasing the number of rAbs is helpful for maximizing the epitope coverage and facilitating high-resolution binning analysis in the practical use of our system. When well-studied antibodies against a particular antigen are limited or unknown, using qAbs as rAbs can provide an alternative option. It is common to utilize qAbs to be analyzed as rAbs, as seen in Fig. A [*MAbs*, 9(1), 29–42 (2017)], with each antibody serving as both the ligand (rAb) and analyte (qAb) in pairwise competitive binding assays. This strategy used in conventional epitope binning methods can also be applied to our Epitope Binning-seq. Some or all of qAbs can be selected and prepared as rAbs for the analysis.

Fig. A. Heat map summarizing the epitope binning results of seven antibodies. Each analyte (qAb) is tested for their binding to the antigen captured by the ligand (rAb) using biolayer interferometry. Green and red cells represent non-blocking and blocking pairs of antibodies, respectively. Cited from *MAbs*, 9(1), 29–42 (2017).

3. Epitope-binning of qAbs was mainly achieved by analyzing the portion of rAb-non-binding cells using flow cytometry. How about the false-positive rate of the binning system?

Answer: Thank you very much for the valuable comment that indicate points for improving our method. In the original manuscript, we utilized the enrichment ratio of 1 as the cutoff to differentiate whether a qAb is enriched or not. Some epitope-dissimilar qAbs showed an enrichment ratio slightly exceeding 1 (highlighted in yellow in Table 1), resulting in false positive rates of 21.4% (3/14 qAbs) and 14.3% (2/14 qAbs), respectively. As discussed in the original manuscript, this may be attributed to the broad setting of the gating region during cell sorting,

which may introduce some undesired cells. This can be addressed by narrowing the sorting region or optimizing other parameters of the cell sorter to ensure entire separation of rAb (-) and rAb (+) populations. Therefore, we have reconsidered the analytical parameters. As a result, we were able to set a relative enrichment ratio of 3 as the cutoff for identifying significantly enriched qAbs. This change allowed 14 qAbs to be accurately classified into their corresponding epitope bins without any false results, demonstrating the increased accuracy of this platform in the binning analysis. We have revised Fig. 6c, 6e and 6f, Supplementary Fig. 6b and 6d, and relating sentences to explain the analysis in the revised manuscript (lines 279-292, 294, 296, 329-331 and 541-544)

(lines 279-292) Upon collecting the rAb(-) cell population that reacted with 0.1 nM Pert-AF647, the percentage of cells displaying Per and its mutants increased significantly (Fig. 6b), with all showing a relative enrichment ratio greater than 3 (Fig. 6c), indicating the significant enrichment of desired qAbs through our epitope binning approach. In contrast, the relative enrichment ratios of Tra and its mutants (which have different epitopes from pertuzumab), and Dac and Niv (which bind to different antigens than HER2), were less than 3. Furthermore, when the reaction concentration of Pert-AF647 was increased to 10 nM, only Per and Per-P53A, which have strong binding affinity, showed a relative enrichment ratio above 3 (Fig. 6d, 6e). When the rAb was changed to 0.1 nM of Tras-AF647, Tra and its mutants exhibited substantial enrichment, all showing the relative enrichment ratio greater than 3, much greater than those of other qAbs (Supplementary Fig. 6a, 6b). As expected, the slight enrichment of scFvs with relatively high binding affinity was observed at the rAb concentrations of 10 nM (Supplementary Fig. 6c, 6d). In this study case, through the identification of significantly enriched antibody clones under the guidance of each rAb at the concentration of 0.1 nM, the 14 qAbs were grouped into three distinct epitope bins: pertuzumab, trastuzumab, and ungrouped (Fig. 6f).

(lines 294 and 296) Specifically, Per and Per mutants, along with Tra and Tra mutants, were classified into the respective pertuzumab- and trastuzumab-epitope groups, which was consistent with their origins, while qAbs Dac and Niv, which exhibited no binding to the antigen, did not belong to either of the two rAb bins.

(lines 329-331) Using Epitope Binning-seq, 14 qAbs were accurately classified into corresponding epitope bins without any false results observed, demonstrating the accuracy of this platform in the binning analysis.

[This sentence has been replaced from the following original discussion: It is worth noting that Per and Niv exhibited an enrichment ratio slightly exceeding 1 (Supplementary Fig. 6d), possibly resulting from the broad setting of the gating region during cell sorting. This can be addressed by narrowing the sorting region or optimizing other parameters of the cell sorter to ensure entire

separation of rAb (-) and rAb (+) populations. Another factor to consider was the high initial abundance of Per and Niv, accounting for 52% and 26% of qAbs, respectively (Supplementary Fig. 6a), implying that an even abundance of each qAb in the initial library may enhance epitope binning accuracy.]

(lines 541-544) Then, the enrichment ratio value of the negative control Niv was set to 1 and the relative enrichment ratio of other qAbs compared to Niv was calculated. When the relative enrichment ratio was greater than 3, it was considered to be significantly enriched.

Table 1. Enrichment ratios of qAbs in the analysis of AF647-Tras

qAbs	0.1 nM	10 nM
Per	0.998	1.106
Per-N52A	1.053	0.436
Per-P53A	0.678	0.076
Per-N54A	1.202	0.039
Per-S55A	1.202	0.068
Per-G56A	0.967	0.114
Tra	5.181	2.913
Tra-G101A	3.489	1.369
Tra-D102A	8.684	2.866
Tra-G103A	6.775	2.371
Tra-F104A	3.666	0.753
Tra-Y105A	5.672	0.484
Dac	0.797	0.817
Niv	0.966	1.428

4. Binding affinity of ScFv for an antigen is known to be quite different from the whole Ab. This might affect the efficiency of the competitive binding assay. Did the authors check the binding affinity of ScFv compared to the whole antibody?

Answer: We thank your important comment. We have measured the binding affinity of whole Abs, pertuzumab and trastuzumab using K562/HER2 cells. The K_D values of pertuzumab and trastuzumab were 0.29 nM and 1.3 nM, respectively (Fig. B). These values were several times lower than their scFvs (Per=9.0 nM and Tra=13 nM) (Fig. 5a). We believe this difference is probably due to the conversion of binding mode from bivalent to monovalent, as seen in previous studies [*PLoS ONE*, 10(4), e0124440 (2015), *Mol Cancer Ther*, 11 (7), 1467–1476 (2012)].

Fig. B. Binding of whole Abs to HER2

K562/HER2 cells were incubated with the Pert-AF647 and Tras-AF647 at concentrations of 0.005, 0.01, 0.05, 0.1, 0.5, 1, 5, 10, and 25 nM for 90 min at 4°C, followed by FCM analysis.

The binding affinity of rAb directly impacts the sensitivity of the competitive binding assay because the competitive pressure is influenced by both the binding affinity and concentration of the rAb. In our system, we have verified that decreasing the concentration of rAb leads to a reduction in competitive pressure and an increase in the detection sensitivity for mutant scFvs with lower affinity (Fig. 5b and 5c, Supplementary Fig. 5a and 5b). If the efficiency of the competitive binding assay is compromised due to a substantial difference in binding affinity between the whole rAb and the scFv qAb, this efficiency can be enhanced by reducing the concentration of the rAb.

Reviewer #4 (Remarks to the Author):

Lin et al. developed Epitope Binning-seq to identify antibodies with similar epitope binding through a competitive binding strategy, employing flow cytometry and DNA sequencing. Specifically, single-chain variable fragments (scFvs), derived from gene transduction, are expressed on the surface of cells that present the antigen. Incubation with fluorescently labeled “reference” antibodies allows for the identification of antigen-presenting cells (APCs) that exhibit no fluorescence. These APCs are presumed to bind to the same epitope as the reference antibodies because the expressed scFv competes for and blocks binding to the antigen. The system was tested using HER2-binding antibodies to evaluate the effects of the antigen and scFv concentrations on sensitivity. Additionally, the ability to identify mutated scFvs with epitope binding similar to that of reference antibodies (rAbs) at various concentrations was assessed. Finally, Epitope Binning-seq was applied to a library of 14 scFvs to determine their epitope binding similarities to the rAb.

Thank you for your valuable comments and suggestions that helped improve our manuscript. The revised words and sentences are colored in blue in the revised manuscript.

The manuscript is well written. I have a few comments below:

1. In Figure 5b and 5c, as well as the corresponding text from lines 252 to 262, it is unclear why the affinities of those specific mutants are compared statistically. Shouldn't the comparison ideally be made with Per? The text mentions: "When cells displaying Per and its mutants were reacted with 0.1 nM Pert-AF647, the percentage of the rAb(-) cell population remained relatively consistent for most scFvs, except for a significant reduction for Per-N52A, which exhibits the lowest binding affinity (Fig. 5a, 5b)." It should be clarified against which scFvs the cell populations are being compared to remain relatively constant.

Answer: Thank you for your valuable comment for improving the clarity of our manuscript. In Fig. 5b, we compared the percentages of rAb(-) cells between Per (52.7%) and its mutants and found that most mutants (except for Per-N52A) showed relatively similar values (49.4–59.0%). In this analysis, statistical comparisons between specific scFvs may confuse the conclusion. Therefore, we have revised the sentence to specify which scFv was used as the reference for the comparison (lines 254-256) and deleted the statistical comparisons in Fig. 5b, 5c and Supplementary Fig. 5a, 5b in the revised manuscript. In addition, we have included a brief conclusion for Fig. 5b (lines 257-258)

(lines 254-256) When cells displaying Per and its mutants were reacted with 0.1 nM Pert-AF647, the percentage of the rAb(-) cell population in Per (52.7%) remained relatively consistent for most Per mutants (49.4–59.0%), except for a significant reduction for Per-N52A (29.3%), which has the lowest binding affinity (Fig. 5a, 5b).

(lines 257-258) These results demonstrate that Per and its mutants bind the same or similar epitopes on HER2.

2. In Figure 3a, the fraction of Pert-AF647-non-binding cells decreases only when the concentration of Pert-AF647 increases to 100 nM, showing no change between 0.1 nM and 10 nM. This contrasts with Figure 5, where at 10 nM, the fraction of Pert-AF647-non-binding cells decreases significantly compared to 0.1 nM, presumably due to lower affinity antibodies. The authors conclude that "These results clearly indicate that the binding affinity of qAbs affects the sensitivity of epitope evaluation, and that lower rAb concentrations allow for the evaluation of both low-affinity and high-affinity qAbs." However, could this also be attributed to the binding of completely different epitopes? Therefore, it raises the question of how similar these Abs are in terms of their epitope binding. Can the epitopes of the different scFv mutations in Figure 5a be predicted using molecular dynamics in a similar manner as the epitopes identified in Figure 2b?

Answer: Thank you for your comment. We believe that there is no contradiction between Fig. 3a and Fig. 5. The percentages of rAb(-) cells in high-affinity qAb, Per, were relatively consistent under all 0.1, 1 and 10 nM rAb conditions (Fig. 3a). The high-affinity qAbs, Per and Per-P53A, exhibited consistent rAb(-) percentages at 0.1 and 10 nM rAb concentrations, while the decrease in percentages of rAb(-) cells at 10 nM rAb concentration was observed in the low-affinity qAbs, such as Per-S55A and Per-N52A (Fig. 5c). These results can be explained that low-affinity qAbs are easier to be displaced by the rAb at a relatively high concentration (10 nM), while high-affinity qAbs can overcome the competitive pressure of high rAb concentration and maintain binding to the antigen. In addition, it's important to note that the presence of rAb(-) cells indicates that the rAb and qAb share a similar epitope. If the displayed qAb binds to a completely different epitope, the rAb(-) cell population cannot be observed at all, as seen in Fig. 2d and 2e. Therefore, we believe that the decrease in the rAb(-) cells in Fig. 5c is attributed to the differences in the affinity of different qAbs rather than binding to completely different epitope.

Determining epitopes by epitope mapping requires detailed information about the antibody-antigen complex structures revealed by high-resolution analyses. In Fig. 2b, we used X-ray crystallographic structures as initial models for MD simulation, obtaining reliable epitope information. In contrast, epitope analysis of their mutants is challenging because of the lack of validated structure information. By computationally substituting residues on scFvs, we can predict their epitopes, but the reliability cannot be guaranteed. Therefore, we have not quantified the epitope similarity among these scFv mutants by MD simulations, and it is difficult to show results that can answer the reviewer's questions in this study. However, we believe that the reviewer question is important in enhancing the versatility of our system, and in future research, we would like to perform epitope mapping that can address this question.

3. Can the authors expand on Figure 6f? Which concentration of Pert-AF647 was used to categorize the scFvs? It appears to have been 0.1 nM. However, at 10 nM, it seems the scFvs would not be grouped correctly. Therefore, would the authors conclude that at 10 nM, Epitope Binning-seq would not group antibodies with similar epitope binding accurately?

Answer: Thank you for your insightful suggestion. We used 0.1 nM rAb for the grouping analysis and have specified the rAb concentration in the revised manuscript (line 292).

In the original manuscript, we utilized the enrichment ratio of 1 as the cutoff to differentiate whether a qAb is enriched or not. Some epitope-dissimilar qAbs showed an enrichment ratio slightly exceeding 1 (highlighted in yellow in Table 1). Therefore, we have reconsidered the analytical parameters, setting a relative enrichment ratio of 3 as the cutoff for identifying significantly enriched qAbs. As a result, we were able to identify significantly enriched qAbs. This change allowed 14 qAbs to be accurately classified into their corresponding epitope bins without any false results, demonstrating the increased accuracy of this platform in the binning analysis.

We have revised Fig. 6c, 6e and 6f, Supplementary Fig. 6b and 6d, and relating sentences to explain the analysis in the revised manuscript (lines 279-292, 294, 296, 329-331 and 541-544)

We believe that changes in rAb concentration would not affect the binning accuracy but rather the detection sensitivity. If we use the results of Epitope Binning-seq using 10 nM Pert-AF647 as rAb (Fig. 6e), only Per and Per-P53A are enriched with a relative enrichment ratio above 3 and there is no false-positive enrichment in other scFvs. This result clearly shows that the detection sensitivity is decreased and the low-affinity qAbs are classified into the ungrouped bin. Therefore, Epitope Binning-seq can still accurately group antibodies with similar epitope at 10 nM rAb, but with a preference toward high-affinity qAbs.

(lines 279-292) Upon collecting the rAb(-) cell population that reacted with 0.1 nM Pert-AF647, the percentage of cells displaying Per and its mutants increased significantly (Fig. 6b), with all showing a relative enrichment ratio greater than 3 (Fig. 6c), indicating the significant enrichment of desired qAbs through our epitope binning approach. In contrast, the relative enrichment ratios of Tra and its mutants (which have different epitopes from pertuzumab), and Dac and Niv (which bind to different antigens than HER2), were less than 3. Furthermore, when the reaction concentration of Pert-AF647 was increased to 10 nM, only Per and Per-P53A, which have strong binding affinity, showed a relative enrichment ratio above 3 (Fig. 6d, 6e). When the rAb was changed to 0.1 nM of Tras-AF647, Tra and its mutants exhibited substantial enrichment, all showing the relative enrichment ratio greater than 3, much greater than those of other qAbs (Supplementary Fig. 6a, 6b). As expected, the slight enrichment of scFvs with relatively high binding affinity was observed at the rAb concentrations of 10 nM (Supplementary Fig. 6c, 6d). In this study case, through the identification of significantly enriched antibody clones under the guidance of each rAb at the concentration of 0.1 nM, the 14 qAbs were grouped into three distinct epitope bins: pertuzumab, trastuzumab, and ungrouped (Fig. 6f).

(lines 294 and 296) Specifically, Per and Per mutants, along with Tra and Tra mutants, were classified into the respective pertuzumab- and trastuzumab-epitope groups, which was consistent with their origins, while qAbs Dac and Niv, which exhibited no binding to the antigen, did not belong to either of the two rAb bins.

(lines 329-331) Using Epitope Binning-seq, 14 qAbs were accurately classified into corresponding epitope bins without any false results observed, demonstrating the accuracy of this platform in the binning analysis.

[This sentence has been replaced from the following original discussion: It is worth noting that Per and Niv exhibited an enrichment ratio slightly exceeding 1 (Supplementary Fig. 6d), possibly resulting from the broad setting of the gating region during cell sorting. This can be addressed by

narrowing the sorting region or optimizing other parameters of the cell sorter to ensure entire separation of rAb (-) and rAb (+) populations. Another factor to consider was the high initial abundance of Per and Niv, accounting for 52% and 26% of qAbs, respectively (Supplementary Fig. 6a), implying that an even abundance of each qAb in the initial library may enhance epitope binning accuracy.]

(lines 541-544) Then, the enrichment ratio value of the negative control Niv was set to 1 and the relative enrichment ratio of other qAbs compared to Niv was calculated. When the relative enrichment ratio was greater than 3, it was considered to be significantly enriched.

Table 1. Enrichment ratios of qAbs in the analysis of AF647-Tras

qAbs	0.1 nM	10 nM
Per	0.998	1.106
Per-N52A	1.053	0.436
Per-P53A	0.678	0.076
Per-N54A	1.202	0.039
Per-S55A	1.202	0.068
Per-G56A	0.967	0.114
Tra	5.181	2.913
Tra-G101A	3.489	1.369
Tra-D102A	8.684	2.866
Tra-G103A	6.775	2.371
Tra-F104A	3.666	0.753
Tra-Y105A	5.672	0.484
Dac	0.797	0.817
Niv	0.966	1.428

Additional revisions

We have made additional revisions as follows:

(lines 32) This versatile platform is applicable to diverse antibodies and antigens, potentially expediting the identification of clinically useful antibodies.

(line 69) Various competitive immunoassay formats (such as classical sandwich, premix, or in-tandem assays) can be used in conjunction with an enzyme-linked immunosorbent assay²⁰, biolayer interferometry, or surface plasmon resonance (SPR)²¹⁻²⁴.

(line 97) We then achieved epitope similarity assessment of four qAb-rAb pairs using dual rAbs with distinct epitopes simultaneously.

(line 214) As a result of the similar emission/excitation wavelengths of AF488 and sfGFP, the qAb cDNA was reconstructed to co-express with blue fluorescent protein (BFP) (Fig. 4a).

(line 224) Furthermore, when the cells displaying different qAbs were mixed pairwise at a 1:1 ratio and evaluated using two rAbs with different epitope recognition, the presence of cells expressing scFv(Per) or scFv(Tra) sharing the same epitope with either Pert-AF647 or Tras-AF488 led to the observation of corresponding populations that exhibited no binding to Pert-AF647 or Tras-AF488 (Fig. 4d).

(line 228) These results suggested that using two distinct rAbs enables the simultaneous evaluation of four rAb-qAb pairs with distinct epitopes in a single experiment.

(line 243) Wild-type and mutant scFvs, scFv(Per), N52A mutant scFv(Per), P53A mutant scFv(Per), N54A mutant scFv(Per), S55A mutant scFv(Per), G56A mutant scFv(Per), scFv(Tra), G101A mutant scFv(Tra), D102A mutant scFv(Tra), G103A mutant scFv(Tra), F104A mutant scFv(Tra), Y105A mutant scFv(Tra), scFv(Dac), and scFv of PD-1 targeting nivolumab [scFv(Niv)] (Supplementary Fig. 2a) are hereafter referred to as Per, Per-N52A, Per-P53A, Per-N54A, Per-S55A, Per-G56A, Tra, Tra-G101A, Tra-D102A, Tra-G103A, Tra-F104A, Tra-Y105A, Dac, and Niv, respectively.

(line 328) In the analysis using Tras-AF647, Tra-G103A, initially representing only 0.018%, was notably enriched (Supplementary Fig. 6a, 6b), indicating the ability to identify and evaluate one clone from a pool of at least 5500 species.

(lines 716, 721, 730, 737, 747, 752, 773, and 777) Sample numbers were added in the corresponding figure legend.

Updating figures

(1) We have replaced the ‘Library cells’ and ‘Sorted cells’ in Figure 6b and Figure 6d with ‘Library clones’ and ‘Sorted clones’, respectively. In addition, we have updated Figure 6c, 6e, 6f, and Supplementary Figure 6b and Supplementary Figure 6d.

Figure 6. Epitope Binning-seq for analyzing various qAbs

Supplementary Figure 6. Epitope Binning-seq for various qAbs using Tras-AF647

(2) We have included individual data points in the plot graphs of the following figures: Figure 2d, 2e, 3a, 3b, 3d, 3e, 5b, and 5c, as well as Supplementary figure 1, 3, 4b and 5. In addition, we have deleted the statistical comparison in Figure 5b and 5c, Supplementary figure 5a and 5b.

Figure 2. Validation of the evaluation system using HER2-binding antibodies

Figure 3. Effects of the Pert-AF647 concentration and the balance between qAb and HER2 expression levels on evaluation sensitivity

Figure 5. Evaluation of qAbs with various HER2-binding affinities

Supplementary Figure 1. Identification of antibody-binding residues on HER2 and HER2-binding residues on antibodies

Supplementary Figure 3. Effects of the Tras-AF647 concentration and the balance between qAb and HER2 expression levels on evaluation sensitivity

b
Supplementary Figure 4. HER2-binding affinity of scFvs

a**b**
Supplementary Figure 5. Evaluation of Tra mutants with various HER2-binding affinities

REVIEWERS' COMMENTS:

Reviewer #2 (Remarks to the Author):

The authors have addressed all of my comments.

Reviewer #3 (Remarks to the Author):

The issues raised by the reviewer have been well addresses, and I recommend publication in the journal.

Reviewer #4 (Remarks to the Author):

The authors have provided comprehensive responses to my comments. I recommend publication.